# Convergence Analysis and Trajectory Comparison of Gradient Descent for Overparameterized Deep Linear Networks

**Hongru Zhao**                                                    *zhao1118@umn.edu*
*School of Statistics*
*University of Minnesota*

**Jinchao Xu**                                                    *jinchao.xu@kaust.edu.sa*
*Computer, Electrical and Mathematical Science and Engineering Division*
*King Abdullah University of Science and Technology*

**Reviewed on OpenReview:** *https: // openreview. net/ forum? id= jG7ndW7UHp*

## Abstract

This paper presents a convergence analysis and trajectory comparison of the gradient descent (GD) method for overparameterized deep linear neural networks with different random initializations, demonstrating that the GD trajectory for these networks closely matches that of the corresponding convex optimization problem. This study touches upon one major open theoretical problem in machine learning–why deep neural networks trained with GD methods are efficient in many practical applications? While the solution of this problem is still beyond the reach of general nonlinear deep neural networks, extensive efforts have been invested in studying relevant questions for deep linear neural networks, and many interesting results have been reported to date. For example, recent results on loss landscape show that even though the loss function of deep linear neural networks is non-convex, every local minimizer is also a global minimizer. We focus on the trajectory of GD when applied to deep linear networks and demonstrate that, with appropriate initialization and sufficient width of the hidden layers, the GD trajectory closely matches that of the corresponding convex optimization problem. This result holds regardless of the depth of the network, providing insight into the efficiency of GD in the training of deep neural networks. Furthermore, we show that the GD trajectory for an overparameterized deep linear network automatically avoids bad saddle points.

## 1 Introduction

Deep linear neural networks, as a class of toy models, are frequently used to understand loss surfaces and gradient-based optimization methods related to non-convex problems. Dauphin et al. (2014) and Choromanska et al. (2015a) explored the loss function of deep non-linear networks based on random matrix theory (such as a spherical spin-glass model). This theory essentially converts the loss surface of deep nonlinear neural networks into that of deep linear neural networks under certain assumptions, some of which are unrealistic. Choromanska et al. (2015b) suggested an open problem to establish a connection between the loss function of neural networks and the Hamiltonian of spherical spin-glass models under milder assumptions. Later, Kawaguchi (2016) successfully discarded most of these assumptions by analyzing the loss surface of the deep linear neural networks.

The landscape for deep linear neural network (Kawaguchi, 2016; Kawaguchi & Lu, 2017; Laurent & Brecht, 2018) focuses on several properties of the critical points: (i) every local minimum is a global minimum; (ii) every critical point that is not a local minimum is a saddle point; and (iii) there exists a saddle such that all eigenvalues of its Hessian are zeros if the network is deeper than three layers. Thus, for deep linear neural networks, convergence to a global minimum is impeded by the existence of poor saddles.

Lee et al. (2016) showed that the gradient method almost surely never converges to a strict saddle point, although the time cost can depend exponentially on the dimension (Du et al., 2017). Gradient descent (GD) with perturbations (Ge et al., 2015; Jin et al., 2017) can find a local minimizer in polynomial time. Thus, the trajectory approach combined with random initialization or random algorithm circumvents the obstacle of existence of poor saddles. According to studies on continuous time dynamics of a gradient flow (Du et al., 2018; Arora et al., 2018b), the balance property of deep linear network is preserved if the initialization is balanced. Arora et al. (2018a;b), Du & Hu (2019), and Hu et al. (2020) successfully proved that GD with its corresponding initialization schemes converges to a global minimizer of deep linear neural networks with high probability. Furthermore, the rate of convergence is linear, and behaves like GD for a convex optimization problem.

Hu et al. (2020) established that the convergence for Gaussian initialization can be very slow for deep linear neural networks with large depths. For efficient convergence of Gaussian networks, the width needs to scale linearly with the depth. They also showed that orthogonal initialization in deep linear neural networks accelerate the convergence. Thus, the convergence behavior of the GD method, for training deep linear neural networks, crucially networks depends on the initialization.

Recent studies have demonstrated the connection between deep learning and kernel methods (Daniely, 2017; Arora et al., 2019a;b; Chizat et al., 2019; Lee et al., 2019; Du et al., 2019; Cao & Gu, 2019; Woodworth et al., 2020), especially the neural tangent kernel (NTK), introduced by Jacot et al. (2018). For most common neural networks, the NTK becomes constant (Jacot et al., 2018; Liu et al., 2020) and remains so throughout the training in the limit of a large layer width. Throughout the training, the neural networks are well described by their first-order Taylor expansion around their parameters at the initialization (Lee et al., 2019).

In this paper, we first study the convergence region, the set of initializations which lead to the linear convergence of GD for deep linear neural networks (see Lemma 3). Next, we demonstrate that if the minimum width among all hidden layers is sufficiently large, then the random initialization will fall into the convergence region with high probability (see Theorem 1, Theorem 4, Theorem 5, and Theorem 6). Furthermore, the worst-case convergence rate of GD for deep linear neural networks is almost the same as the original convex optimization problem with a corresponding learning rate. The most significant finding of our work is that the trajectories of gradient descent for deep linear neural networks are arbitrarily close to those of the corresponding convex optimization problem, establishing a strong connection between their optimization dynamics and providing insights into gradient descent behavior in more complex neural networks. The precise statement is related to Remark 3, Theorem 2, Lemma 6, and Corollary 3.

## 1.1 Our contributions

This work is inspired by recent work by Du & Hu (2019) and Hu et al. (2020), who carefully constructed the upper and lower bounds of eigenvalues of the Gram matrix along GD and established linear convergence. In this paper, we generalize their results to strongly convex loss functions with layer-varying widths, and obtain sharp results. We show that our rate of convergence for GD in deep linear neural networks matches the worst-case convergence rate for the original convex optimization problem. Next, we show that the optimal rate depends neither on the types of random initialization nor on the depth of the network, provided that the width of each hidden layer is appropriately large.

The most significant contribution of our work is the trajectory comparison result, which demonstrates that the trajectories of gradient descent for deep linear neural networks can be made arbitrarily close to those of the original convex optimization problem (1). To the best of our knowledge, this trajectory comparison result is the first of its kind and has the potential to greatly influence our understanding of optimization in deep learning.

## 2 Preliminaries

### 2.1 Problem Setup

Let $x \in \mathbb{R}^{n_x}$ and $y \in \mathbb{R}^{n_y}$ be an input vector and a target vector, respectively. Define $\{(x_i, y_i)\}_{i=1}^{m}$ as a training dataset of size $m$, and let $X = [x_1, x_2, \cdots, x_m] \neq 0$ and $Y = [y_1, y_2, \cdots, y_m]$. Denote the weight parameters by $W \in \mathbb{R}^{n_y \times n_x}$.

Consider the well studied convex optimization problem:

$$\underset{W}{\text{minimize}} \quad L(W) := \frac{1}{m} \sum_{i=1}^{m} l(Wx_i, y_i), \tag{1}$$

where $l(\cdot, y)$ is strongly convex and has a smooth gradient with respect to its first argument uniformly in $y$.

The GD for convex optimization problem (1) with learning rate $\eta_*$ is given by:

$$W(t+1) = W(t) - \eta_* \nabla L(W(t)), t = 0, 1, 2, \cdots . \tag{2}$$

For any matrix $A$, let $\sigma_{max}(A)$ and $\sigma_{min}(A)$ be the largest and smallest singular values of $A$ respectively.

In this paper, we consider two types of matrix norms and one type of semi-norm for $A$, $\|A\| := \sigma_{max}(A), \|A\|_F^2 := \text{tr}(AA^T)$, and $\|A\|_X := \|AP_X\|_F$, where $P_X = X(X^T X)^\dagger X^T$ is the orthogonal projection matrix onto the column space of $X$, and $(X^T X)^\dagger$ is the Moore–Penrose inverse.

For two real matrices $A, B$ with the same size, we consider their Frobenius inner product as well as their semi-inner product, $\langle A, B \rangle = \langle A, B \rangle_F := \text{tr}(A^T B)$ and $\langle A, B \rangle_X := \langle AP_X, BP_X \rangle$.

The following lemma illustrates why semi-norm $\|\cdot\|_X$ is crucial in our analysis.

**Lemma 1** *Assume that $l(\cdot, y)$ is $\alpha(l)-$strongly convex. The following statements hold.*

1. *If $X$ is not full row rank, then $L(W)$ is neither strictly convex nor strongly convex with respect to $\|\cdot\|_F$.*

2. *$L(W)$ is $\frac{\alpha(l)\lambda_{min}(XX^T)}{m}-$strongly convex with respect to $\|\cdot\|_X$, where $\lambda_{min}(XX^T)$ is the smallest non-zero eigenvalue of $XX^T$.*

The proof of the lemma above can be found in Appendix A. From now on, if we do not specify the inner product, we will always consider the semi-inner product.

Assume that $L$ is $\alpha-$strongly convex ($\alpha > 0$), and $\nabla L$ is $\beta-$Lipschitz (with respect to semi-norm $\|\cdot\|_X$), that means, for any $W, V \in \mathbb{R}^{n_y \times n_x}$,

$$L(W) \geq L(V) + \langle \nabla L(V), W - V \rangle_X + \frac{\alpha}{2} \|W - V\|_X^2,$$

$$\|\nabla L(W) - \nabla L(V)\|_X = \|\nabla L(W) - \nabla L(V)\|_F \leq \beta \|W - V\|_X.$$

Without loss of generality, we assume that $\alpha$ and $\beta$ are the best constants. Then, Lemma 1 implies $\alpha \geq \frac{\alpha(l)\lambda_{min}(XX^T)}{m}$. Similarly, we can also show that $\beta \leq \frac{\beta(l)\lambda_{max}(XX^T)}{m}$, where $\nabla l(\cdot, y)$ is $\beta(l)-$Lipschitz and $\lambda_{max}(XX^T)$ is the largest eigenvalue of $XX^T$.

Define the effective condition number of the convex function $L$ by $\kappa = \kappa(L) = \frac{\beta}{\alpha} < \infty$. $\kappa$ appears naturally in the rate of convergence of GD. Let $W_*$ be a global minimizer of $L(W)$, that is, $L(W_*) = \min_W L(W)$. Notice that $W_*$ might not be unique, but $W_* P_X$ is unique, that is, if $W'$ is another global minimizer of $L(W)$, then $W_* P_X = W' P_X$ (see Lemma 7 for the precise statement).

By applying the well-known results on the rate of convergence, specifically Theorem 3.9 from Garrigos & Gower (2023) and Theorem 2.1.15 from Nesterov (2003), for GD (2), we have

$$\eta_* = \frac{1}{\beta} \implies \mathcal{E}(t) \leq \left(1 - \frac{1}{\kappa}\right)^t \mathcal{E}(0), t = 1, 2, \cdots, \text{ as well as,} \tag{3}$$

$$\eta_* = \frac{2}{\alpha + \beta} \implies \mathcal{E}(t) \leq \frac{\beta}{2}\left(1 - \frac{4\kappa}{(1+\kappa)^2}\right)^t \|W(0) - W_*\|_X^2, t = 1, 2, \cdots, \tag{4}$$

where $\mathcal{E}(t) = L(W(t)) - L(W_*)$.

## 2.2 Deep Linear Network Setup

Let $N - 1$ be the number of hidden layers. Assume $rank(X) = r$. Denote the weight parameters by $W_k \in \mathbb{R}^{n_k \times n_{k-1}}, k = 1, 2 \cdots, N$, with $n_N = n_y, n_0 = n_x$, where the $n_k$ is the width of the $k$-th layer. Set $n_{min} = \min\{n_1, n_2, \cdots, n_{N-1}\}$ and $n_{max} = \max\{n_1, n_2, \cdots, n_{N-1}\}$. For notational convenience, we denote $n_{j:i} = \prod_{i \leq k \leq j} n_k$ and denote $W_{j:i} = W_j W_{j-1} \cdots W_i$ for each $1 \leq i \leq j \leq N$. Define $n_{i-1:i} = 1$ and $W_{i-1:i} = I$ (of appropriate dimension) for completeness.

Consider the implicit regularization on $W = W_{N:1}$ for the convex optimization problem (1). Thus, we obtain the following non-convex optimization problem of deep linear neural networks:

$$\underset{W_1, \cdots, W_N}{\text{minimize}} \quad L(W_{N:1}) = \frac{1}{m} \sum_{i=1}^m l(W_{N:1} x_i, y_i). \tag{5}$$

**Example 1** *Specifically, if we set the loss to be $l(W x_i, y_i) = \|W x_i - y_i\|_2^2$, then $L(W) = \frac{1}{m}\|WX - Y\|_F^2$ is $\frac{2\lambda_{min}(XX^T)}{m}$-strongly convex, and $\nabla L$ is $\frac{2\lambda_{max}(XX^T)}{m}$-Lipschitz.*

**Example 2** *The deep linear neural networks with regularization $\lambda \|W_N \cdots W_1 P_X\|_F^2$ can be converted into a new optimization problem*

$$\underset{W_1, \cdots, W_N}{\text{minimize}} \quad L(W_{N:1}) + \lambda \|W_{N:1}\|_X^2.$$

*Let $L_\lambda(W) = L(W) + \lambda \|W\|_X^2$. Then $L_\lambda(\cdot)$ is $\alpha + 2\lambda$-strongly convex, and $\nabla L_\lambda(\cdot)$ is $\beta + 2\lambda$-Lipschitz.*

*More general, if we consider regularization with form $R(W) = \lambda \cdot g(W P_X)$, $g(\cdot)$ is $\alpha'$-strongly convex, and $\beta'$-Lipschitz, then for the optimization problem*

$$\underset{W_1, \cdots, W_N}{\text{minimize}} \quad L(W_{N:1}) + R(W_N \cdots W_1) =: L_R(W_N \cdots W_1),$$

*we know that $L_R(\cdot)$ is $\alpha + \lambda\alpha'$-strongly convex, and $\nabla L_R(\cdot)$ is $\beta + \lambda\beta'$-Lipschitz.*

## 2.3 Initialization Schemes

In literature, researchers consider the following form of the deep linear network instead of (5):

$$\underset{W_1, \cdots, W_N}{\text{minimize}} \quad L(a_N W_{N:1}) = \frac{1}{m} \sum_{i=1}^m l(a_N W_{N:1} x_i, y_i), \tag{6}$$

where $a_N = 1/\sqrt{n_1 n_2 \cdots n_N}$ is a normalization constant.

Applying GD on (6), where we update $W_j$ simultaneously for $j$,

$$W_j(t+1) = W_j(t) - \eta \cdot a_N \left(W_{N:j+1}(t)\right)^T \nabla L\left(a_N W_{N:1}(t)\right)\left(W_{j-1:1}(t)\right)^T, j = 1, \cdots, N. \tag{7}$$

Recent work consider GD (7), initializing $W_j(0)$ with Gaussian initialization (Du & Hu, 2019) or scaled orthogonal initialization (Hu et al., 2020).

In this paper, we consider the following three kinds of random initialization which generalizes their idea.

● **Gaussian Initialization:** Let $W_1(0), \cdots, W_N(0)$ be the weight matrices at initialization. We assume that all entries of $W_j, 1 \leq j \leq N$ are independent Gaussian random variables with zero mean and unit variance.

Then $a_N$ is a normalization constant in the sense that for any $x \in \mathbb{R}^{n_0}$, we have

$$\mathbb{E}\left[\|a_N W_{N:1}(0)x\|_2^2\right] = \|x\|_2^2. \tag{8}$$

In fact, all initializations we discussed in this paper satisfy (8); see Lemma 9, Lemma 14 and the proof of Theorem 6.

**Remark 1** *Let $V_i(t) = \frac{1}{\sqrt{n_i}}W_i(t)$, for $1 \leq i \leq N$. Then, GD (7) with unit variance Gaussian initialization is equivalent to*

$$V_j(t+1) = V_j(t) - \frac{\eta}{n_j}\left(V_{N:j+1}(t)\right)^T \nabla L\left(V_{N:1}(t)\right)\left(V_{j-1:1}(t)\right)^T, \tag{9}$$

*with mean zero and variance $\frac{1}{n_i}$ Gaussian initialization for $V_i, i = 1, \cdots, N$. The GD (9) for the loss (5) is equivalent to GD (7) for the loss (6). From now on, we will only consider GD (7) for deep linear neural networks (6).*

● **Orthogonal Initialization:** We consider a so-called one peak random orthogonal projections and embeddings initialization, which generalizes the idea of orthogonal initialization (Hu et al., 2020).

**Definition 1** *A initialization $W_{N:1}(0) = W_N(0)W_{N-1}(0)\cdots W_1(0)$ is said to be one peak random orthogonal projections and embeddings initialization if there exists $1 \leq p < N$, such that $n_0 \leq n_1 \leq n_2 \leq \cdots \leq n_p$, $n_p \geq n_{p+1} \geq n_{p+2} \geq \cdots n_{N-1} \geq n_N$, and $W_1(0), W_2(0), \cdots, W_p(0), W_{p+1}(0), W_{p+2}(0), \cdots, W_{n_N}(0)$ are independent and uniformly distributed over rectangular matrices which satisfy*

$$\begin{cases} W_i^T(0)W_i(0) = n_i I_{n_{i-1}}, 1 \leq i \leq p, \\ W_j(0)W_j^T(0) = n_{j-1}I_{n_j}, p+1 \leq j \leq N. \end{cases}$$

**Remark 2** *In this definition, $\sqrt{\frac{1}{n_i}}W_i(0), 1 \leq i \leq p$ are random embeddings and $\sqrt{\frac{1}{n_{j-1}}}W_j(0), p+1 \leq j \leq N$ are random orthogonal projections. Notice that, $A$ is a random orthogonal projection if and only if $A^T$ is a random embedding.*

Arora et al. (2018a) studied the rate of convergence to global optimum for GD to train deep linear neural network for balanced initialization. Here, we will consider a special case of balanced initialization which is described as follows.

● **Special Balanced Initialization:** Assume $n_1 = \cdots = n_{N-1} = n$. Consider initialization $W_N(0) = \sqrt{n}U_N[I_{n_y}, 0_{n_y \times (n-n_y)}]V_N^T$, $W_1(0) = \sqrt{n}U_1[I_{n_x}, 0_{n_x \times (n-n_x)}]^T V_1^T$ and $W_i(0) = \sqrt{n}U_i I_n V_i^T, 2 \leq i \leq N-1$, where $U_{N-1}, U_N, V_1, V_i = U_{i-1}, 2 \leq i \leq N-1$ are orthogonal matrices (random or deterministic), and $V_N$ has uniform distribution over orthogonal matrices. Notice that only $V_N$ is required to be random.

A simple estimation of the loss at the initialization is given by the following lemma.

**Lemma 2** *If the initialization satisfies (8) for all $x$, then with probability at least $1 - \frac{\delta}{2}$, we have*

$$L(a_N W_{N:1}(0)) - L(W_*) \leq \beta B_\delta, \ \ where \ B_\delta = \left(\frac{2 \cdot rank(X)}{\delta} + \|W_*\|_X^2\right).$$

Note that using sharp concentration inequality, the bound $B_\delta$ can be improved.

# 3 Main Results

For the rest of this paper, assume the thinnest layer is either the input layer or the output layer, that is, $n_{min} \geq \max\{n_0, n_N\}$ and the ratio between the width of any hidden layer is bounded from above, precisely we have $\frac{n_{max}}{n_{min}} \leq C_0 < \infty$. Define some quantities as follows:

$$q = \begin{cases} 1 - \alpha\eta_*(2 - \eta_*\alpha), & 0 < \eta_* \leq \frac{2}{\alpha+\beta}, \\ 1 - \beta\eta_*(2 - \eta_*\beta), & \frac{2}{\alpha+\beta} < \eta_* < \frac{2}{\beta}, \end{cases} \tag{10}$$

$$\begin{aligned} \mathbb{C}_1 &= n_N\kappa^2 B_\delta C_0 + \ln N, \\ \mathbb{C}_2 &= n_N\kappa^2 B_\delta C_0 + C_0 \ln(\underline{N}), \\ \mathbb{C}_3 &= n_N\kappa^2 B_\delta, \end{aligned} \tag{11}$$

where $\underline{N}$ denotes the number of distinct elements in the set $\{n_1, \cdots, n_{N-1}\}$.

Here, the quantity $q$ is related to the rate of convergence of gradient descent for strongly convex function $L(W)$. Setting the learning rate $\eta_* = \frac{2}{\alpha+\beta}$, we can rewrite the well-known rate in (4) in terms of $q$, due to

$$q = 1 - \alpha\eta_*(2 - \eta_*\alpha) = 1 - \frac{4\kappa}{(1+\kappa)^2}.$$

The parameter $\delta \in (0, \frac{1}{2})$ is a confidence threshold, such that the conclusions in Theorem 1 (or Theorem 2) hold with a probability of at least $1 - \delta$, given an extra inequality constraint between the minimum width of hidden layers, $n_{min}$, and the depth, $N$, as quantified by $\mathbb{C}_1$, $\mathbb{C}_2$, and $\mathbb{C}_3$.

For notational convenience, we denote

$$\mathcal{E}(t) = L(W(t)) - L(W_*), \text{ and } \mathcal{E}_{DLN}(t) = L(a_N W_{N:1}(t)) - L(W_*).$$

Our assumptions and notation are now in place. We next state our main theorems in this section.

## 3.1 Linear Convergence of Deep Linear Neural Networks

In Appendix B, we present an analysis of the linear convergence of GD for deep linear neural networks, in Theorem 4 for Gaussian initialization, Theorem 5 for orthogonal initialization, and Theorem 6 for special balanced initialization, respectively. These results extend the work of Du & Hu (2019), and Hu et al. (2020), which proved similar convergence rates with $\ell_2$ loss. In particular, with the special choice of learning rate $\eta = \frac{n_N}{\beta N}$, our Theorem 4 and Theorem 5 lead us to obtain the following optimum convergence rate.

**Theorem 1** *Consider GD for the deep linear neural networks (7) with learning rate $\eta = \frac{n_N}{\beta N}$. Given any $\delta, \varepsilon \in (0, \frac{1}{2})$ and $C_0 \in [1, \infty)$, there exists a constant $C := C(\varepsilon)$, such that if one of the following two overparameterization condition holds:*

*1. $n_{min} \geq C \cdot \mathbb{C}_1 \cdot N$ with the Gaussian initialization,*

*2. $n_{min} \geq C \cdot \mathbb{C}_2$ with the one peak random orthogonal projections and embeddings initialization,*

*with probability at least $1 - \delta$ we have*

$$\mathcal{E}_{DLN}(t) \leq \left(1 - \frac{1-\varepsilon}{\kappa}\right)^t \mathcal{E}_{DLN}(0), t = 1, 2, \cdots.$$

**Remark 3** *Consider GD (2) with learning rate $\eta_* = \frac{1}{\beta}$, and initialization $W(0) = a_N W_{N:1}(0)$. The well-known result on the rate of convergence (3) for GD (2) of the convex optimization problem (1) matches the rates in Theorem 1.*

**Remark 4** *If we set $C_0 = 1$ and consider $\ell_2$ loss, then we can recover the main results in Du & Hu (2019), and Hu et al. (2020), that the number of iterations needed to reach precision $\varepsilon$ is $O\left(\kappa \log \frac{1}{\epsilon}\right)$ for $\ell_2$ loss. We generalized their results to any strongly convex loss with varying width.*

### 3.2 Trajectory Comparison

Theorem 1 and Remark 3 establish that the rate of convergence to a global optimum for GD to train a deep linear neural networks is almost the same as the trajectories for GD to train the corresponding convex optimization problem with high probability, if the width is large enough. Moreover, GD for the fully-connected deep linear neural networks (7) and that for GD (2) have almost the same trajectories.

Let $\eta_1 = \frac{2n_N}{\beta N}$ be an upper bound of learning rate $\eta$. We can show that the trajectories of GD (7) for deep linear neural networks (6) with learning rate $\eta < \eta_1$ are close to those of GD (2) with learning rate $\eta_* = \frac{N}{n_N}\eta$ for the corresponding convex optimization problem (1) with high probability, if the width of each hidden layer is sufficiently large. The precise statement is as follows.

**Theorem 2 (Trajectory Comparison)** *Consider GD for the deep linear neural networks (7) with learning rate $\eta < \eta_1$ for $a_N W_{N:1}(t)$, $t = 0, 1, \cdots$, and GD (2) with learning rate $\eta_* = \frac{N}{n_N}\eta$ for $W(t)$, $t = 0, 1, \cdots$. Given $\tau, \delta \in (0, 1)$ and $C_0 \in [1, \infty)$, there exists a constant $C := C(\tau, \eta/\eta_1)$ such that if one of the following three overparameterization conditions holds:*

1. *$n_{min} \geq C \cdot \mathbb{C}_1 \cdot N$ with the Gaussian initialization,*

2. *$n_{min} \geq C \cdot \mathbb{C}_2$ with the one peak random orthogonal projections and embeddings initialization,*

3. *$n_{min} \geq C \cdot \mathbb{C}_3$ with the special balanced initialization,*

*then with probability at least $1 - \delta$, we have*

$$\|a_N W_{N:1}(t) - W(t)\|_X^2 \leq D(\tau, q, t) \|a_N W_{N:1}(0) - W_*\|_X^2, \tag{12a}$$

$$|\mathcal{E}_{DLN}(t) - \mathcal{E}(t)| \leq \beta \left( q^{t/2}\sqrt{D(\tau, q, t)} + \frac{1}{2}D(\tau, q, t) \right) \|a_N W_{N:1}(0) - W_*\|_X^2, \tag{12b}$$

$$\mathcal{E}_{DLN}(t) \leq 3\beta(q + \tau)^t \|a_N W_{N:1}(0) - W_*\|_X^2, \tag{12c}$$

*where $D(\tau, q, t) = \min\left\{ \frac{\tau}{1-q}, 2(q + \tau)^t \right\}$, with $0 < q < 1$ defined in (10).*

**Remark 5** *To our knowledge, we are the first who showed that the trajectory of the overparameterized deep linear neural networks is close to the original convex optimization problem with appropriately rescaled learning rate.*

**Corollary 3** *Under the setting of Theorem 2, if we set $\eta = \frac{2n_N}{(\alpha+\beta)N}$, the following inequality hold with high probability,*

$$\mathcal{E}_{DLN}(t) \leq 3\beta \left( 1 - \frac{4\kappa}{(1+\kappa)^2} + \tau \right)^t \|a_N W_{N:1}(0) - W_*\|_X^2. \tag{13}$$

*We notice that the rate of convergence in (13), matching (4), is better than that in Theorem 1. It is because if $\kappa > 1$, we can choose $\tau$ sufficiently small, so that the following inequality holds:*

$$1 - \frac{4\kappa}{(1+\kappa)^2} + \tau < 1 - \frac{1}{\kappa}.$$

Roughly speaking, Theorem 1, Theorem 2, Theorem 4, together with Theorem 5 illustrate that the GD for a convex optimization problem recovers the convex problem itself in terms of optimization, at the cost of linear convergence only with high probability for random initialization.

**Remark 6** *In constants $\mathbb{C}_1, \mathbb{C}_2, \mathbb{C}_3$ defined in (11), the term $\frac{rank(X)}{\delta}$ is not optimal, since our concentration inequality only depends on the second moment. By using stronger concentration inequalities for our Lemma 2, similar to the proof of proposition 6.5 (Du & Hu, 2019) and Lemma 4.2 (Hu et al., 2020), the $\frac{rank(X)}{\delta}$ can be improved to $1 + \log(\frac{rank(X)}{\delta})$.*
*The $\mathbb{C}_1$ is proportional to $\kappa^2$, which is slightly better than the constant in Du & Hu (2019) that is proportional to $\kappa^3$. The $\mathbb{C}_2$ is also slightly better than the constant in Hu et al. (2020), since we do not have the extra term $\frac{\|X\|_F^2}{\|X\|^2}$. The improvement of the constant is mainly due to introducing the semi-norm $\|\cdot\|_X$.*

### 3.3 Proofs Overview

Now we will give an overview of the proofs for all theorems in the main results. Since Theorem 1 in the main results are special cases of general theorems with non-optimal learning rate (see Theorem 4 and Theorem 5), we only need to focus on the proofs of the general theorems (see Theorem 2, Theorem 4, Theorem 5, and Theorem 6).

We start with the convergence region of deep linear neural networks. Basically, the convergence region is the set of initialization which lead to the convergence of GD for the deep linear neural networks. The precise definition can be found in the Definition 2. Lemma 3 and Lemma 6 prove that this convergence region satisfies the following properties: if the initialization falls into the convergence region, then

(i) GD is guaranteed to converge to a global minimizer of the deep linear neural networks,

(ii) the worst-case rate of convergence of GD for the deep linear neural networks, which is a non-convex problem, is almost the same as the corresponding convex optimization problem with a corresponding learning rate, and,

(iii) the trajectories of GD for the deep linear neural networks are arbitrarily close to those for the corresponding convex optimization problem.

More precisely, Lemma 3 establishes the convergence region for deterministic initialization, and it demonstrates the first two properties, (i) and (ii). Additionally, in Appendix E and Appendix F we also proved that the spectral properties of products of random matrices partially reveal the mystery of overparameterization, that is, overparameterization by adding width of each hidden layer guarantees that the random initialization will fall into the convergence region with high probability. These results provide a foundation to establish the main linear convergence theorem for random initialization (see Theorem 4, 5, and 6).

On the other hand, Lemma 4 shows that if the initialization falls into the convergence region, the update rule for the product of weight matrices in GD for deep linear neural networks is more or less that given in (2). This result can be used to establish both Theorem 2, and Lemma 6, which is precisely the property (iii) of the convergence region for deterministic initialization and non-deterministic initialization, respectively. It is worth mentioning that property (iii) of the convergence region cannot be directly obtained from the results of Du & Hu (2019) and Hu et al. (2020), highlighting the contribution of our work in establishing this property.

## 4 Convergence Analysis and Trajectory Comparison

### 4.1 Initialization and Convergence Region

Arora et al. (2018a) showed that if the initialization is approximately balanced, and the product matrix $W_{N:1}(0)$ is very close to a global minimizer, then GD linearly converges to the global minimum for the deep linear network, without any requirement on the width. However, the convergence region in (Arora et al., 2018a) is very small, since $W_{N:1}(0)$ has to be very close to $W_*$. Later, several papers by Du & Hu (2019), and Hu et al. (2020) successfully proved that GD with Gaussian, or orthogonal initialization linearly converges to a global minimizer of the overparameterized deep linear neural networks with high probability. They introduced a technique to analyze trajectories of GD with large widths for any deterministic initialization.

We introduce the following lemma which generalizes the idea from recent work (Du & Hu, 2019; Hu et al., 2020), describes the linear convergence result for deep linear networks with deterministic initialization.

Define $A|_{\mathcal{R}(X)} = AX^T(XX^T)^- X = AP_X$, and view $A|_{\mathcal{R}(X)}$ as a linear operator on $\mathcal{R}(X)$, the column space of $X$. For notational convenience, we denote $W_{j:i}(t) = W_j(t) \cdots W_i(t)$, $L_t = L(a_N W_{N:1}(t))$, and $\nabla L_t = \nabla L(a_N W_{N:1}(t))$, etc.

**Lemma 3** *Assume the initialization satisfies the following conditions simultaneously:*

$$
\begin{cases}
\sigma_{max}(W_{N:i+1}(0)) \le e^{c_1/2}(n_{N-1:i})^{1/2}, 1 \le i \le N-1, \\
\sigma_{min}(W_{N:i+1}(0)) \ge e^{-c_2/2}(n_{N-1:i})^{1/2}, 1 \le i \le N-1, \\
\sigma_{max}(W_{i-1:1}(0)|_{\mathcal{R}(X)}) \le e^{c_1/2}(n_{i-1:1})^{1/2}, 2 \le i \le N, \\
\sigma_{min}(W_{i-1:1}(0)|_{\mathcal{R}(X)}) \ge e^{-c_2/2}(n_{i-1:1})^{1/2}, 2 \le i \le N, \\
\|W_{j:i}(0)\| \le M/2 \cdot N^\theta (\prod_{i \le k \le j-1} n_k \cdot \max\{n_{i-1}, n_j\})^{1/2}, 1 < i \le j < N, \\
L_0 - L(W_*) \le \beta B_0 =: B,
\end{cases}
\tag{14}
$$

*where $c_1, c_2, M$ are positive constant and $\theta \ge 0$. Note that $B_0$ is a proper upper bound for $\|a_N W_{N:1}(0)\|_X^2 + \|W_*\|_X^2$.*

*Set the learning rate $\eta = \frac{(1-\varepsilon)2n_N}{e^{6c_1+3c_2}\beta N}$, where $0 < \varepsilon < 1$. Define $\gamma = \frac{2e^{6c_1}\varepsilon\alpha N}{n_N}$. Assume that*

$$
n_{min} \ge \frac{C(c_1,c_2)M^2\kappa^2 B_0}{\varepsilon^2} N^{2\theta} n_N. \tag{15}
$$

*Then GD (7) satisfies*

$$
L_t - L(W_*) \le (1-\eta\gamma)^t (L_0 - L(W_*)), t = 1, 2, \cdots.
$$

**Definition 2** *For given $c_1, c_2, M, B_0 > 0$, and $\theta \ge 0$, we define the convergence region $\mathfrak{R}(c_1, c_2, \theta, M, B_0)$ by the set of initialization that satisfies the inequality system (14).*

**Remark 7** *Condition (14) describes the convergence region for initialization and the condition (15) describes the overparameterization for deep linear neural networks. At this time, it is not clear how large this convergence region is. Later, we will show that the properly scaled random initialization with some extra mild overparameterization conditions will fall into this convergence region with high probability.*

Our convergence region (see (14) in Lemma 3 and Definition 2) originated from Du & Hu (2019), and Hu et al. (2020). This convergence region can be view as a neighbourhood of special balanced initialization, if $n_1 = n_2 = \cdots = n_{N-1}$. Both the Gaussian and orthogonal initializations are approximately balanced.

For the $\ell_2$ loss, we assume without loss of generality that $X$ is of full rank and $L(W_*) = 0$, due to the decomposition method in Claim B.1 from Du & Hu (2019). However, when considering the general strongly convex loss, we have to directly confront the low rank $X$ in our analysis. Thus, the $\|\cdot\|_X$ appears naturally and helps us to achieve the sharp rate of convergence in our main theorems.

## 4.2 Dynamic of GD for Overparameterized Deep Linear Neural Networks

We will explain why trajectories of GD for overparameterized deep linear neural networks with approximate balanced initialization are close to trajectories for convex problems. Although the recent result (Ziyin et al., 2022) can describe the exact global minimizer for the deep linear network (with a regularization term such as $\ell_2$), the evolution of each $W_j$ is still hard to track. Instead, we consider the discrete dynamics for product matrices $W_{N:1}(t)$:

$$
a_N W_{N:1}(t+1) = a_N W_{N:1}(t) - \eta \cdot P(t)[\nabla L(a_N W_{N:1}(t)P_X)] + a_N E(t),
$$

where linear operator

$$
P(t)[A] = a_N^2 \sum_{i=1}^{N} W_{N:i+1}(t)W_{N:i+1}^T(t)(AP_X)(W_{i-1:1}(t)|_{\mathcal{R}(X)})^T W_{i-1:1}(t)|_{\mathcal{R}(X)},
$$

for any $A \in \mathbb{R}^{n_N \times n_0}$.

Du & Hu (2019) showed for their linear operator $P_t$ that $\lambda_{max}(P_t) \le O(\frac{N}{n_N}) \cdot \lambda_{max}(X^T X)$ and $\lambda_{min}(P_t) \ge \Omega(\frac{N}{n_N}) \cdot \lambda_{min}(X^T X)$. We prove that for our operator $P(t)[\cdot] \approx \frac{N}{n_N} I$ (also see (43)), where $I$ is the identity operator. $E(t)$ is negligible, which leads to the following lemma on discrete dynamics.

### 4.3 Trajectory Comparison

We can summarize the above result in the following dynamic comparison lemma.

**Lemma 4** *Assume that all the assumptions in Lemma 3 hold. For any $\tau > 0$, we can choose new constants $c_1, c_2$ as well as $C := C(c_1, c_2)$ such that the overparameterization assumption (15) in Lemma 3 hold and*

$$\|R(t)\|_X \leq \tau \|a_N W_{N:1}(t) - W_*\|_X, \tag{16}$$

*where*

$$a_N W_{N:1}(t+1) = a_N W_{N:1}(t) - \frac{N}{n_N} \eta \nabla L(a_N W_{N:1}(t)) + R(t).$$

Without the $R(t)$ term, the discrete dynamics is exactly GD (2) for a convex function (1). To control the distance between the two trajectories, we introduce the following lemma, which coincided with Theorem 6 in Hacohen & Weinshall (2022).

Next, we introduce the following lemma for convergence analysis for a dynamical system $V(t+1) = V(t) - \eta_* \nabla L(V(t)) + R(t)$, with $V(t) = a_N W_{N:1}(t)$, $t = 0, 1, \cdots$.

**Lemma 5** *Consider a discrete dynamical system $V(t)$ such that,*

$$V(t+1) = V(t) - \eta_* \nabla L(V(t)) + R(t), t \geq 0,$$

*where $\|R(t)\|_X \leq \tau \|V(t) - W_*\|_X$, and $\tau \in [0,1)$. If $\eta_* \leq 2/\beta$, we have*

$$\|V(t) - W_*\|_X^2 \leq (q + 7\tau)^t \|V(0) - W_*\|_X^2,$$

*where $0 < q < 1$ is defined in (10).*

**Proof of Lemma 5**  Set $\Delta(t) = V(t) - W_*$ and $\tau' = \tau \|\Delta(t)\|_X$. Notice that

$$\Delta(t+1) = \Delta(t) - \eta_*(\nabla L(V(t)) - \nabla L(W_*)) + R(t),$$

and

$$\|\Delta(t+1)\|_X^2 \leq \eta_*^2 \|\nabla L(V(t)) - \nabla L(W_*)\|_X^2 - 2\eta_* \langle \Delta(t), \nabla L(V(t)) - \nabla L(W_*) \rangle_X$$
$$+ \|\Delta(t)\|_X^2 + (2\|\Delta(t)\|_X + 2\eta_* \|\nabla L(V(t)) - \nabla L(W_*)\|_X + \tau')\tau'.$$

By inequality (28) in Appendix C, we have

$$\|\Delta(t+1)\|_X^2 \leq \|\Delta(t)\|_X^2 - 2\eta_* \langle \Delta(t), \nabla L(V(t)) - \nabla L(W_*) \rangle_X$$
$$+ \eta_*^2 \|\nabla L(V(t)) - \nabla L(W_*)\|_X^2 + 7\tau \|\Delta(t)\|_X^2$$
$$= (1 + 7\tau) \|\Delta(t)\|_X^2 - 2\eta_* \langle \Delta(t), \nabla L(V(t)) - \nabla L(W_*) \rangle_X$$
$$+ \eta_*^2 \|\nabla L(V(t)) - \nabla L(W_*)\|_X^2$$
$$\leq (1 + 7\tau) \|\Delta(t)\|_X^2 - 2\eta_* \frac{\alpha\beta}{\alpha + \beta} \|\Delta(t)\|_X^2$$
$$+ \left( \eta_*^2 - \frac{2\eta_*}{\alpha + \beta} \right) \|\nabla L(V(t)) - \nabla L(W_*)\|_X^2.$$

**Case 1**: $\frac{2}{\alpha + \beta} < \eta_* < \frac{2}{\beta}$.
In this case, we have

$$\|\Delta(t+1)\|_X^2 \leq (1 + 7\tau) \|\Delta(t)\|_X^2 - 2\eta_* \frac{\alpha\beta}{\alpha + \beta} \|\Delta(t)\|_X^2 + \left( \eta_*^2 - \frac{2\eta_*}{\alpha + \beta} \right) \|\nabla L(V(t)) - \nabla L(W_*)\|_X^2$$
$$\leq (1 + 7\tau) \|\Delta(t)\|_X^2 - 2\eta_* \frac{\alpha\beta}{\alpha + \beta} \|\Delta(t)\|_X^2 + \left( \eta_*^2 - \frac{2\eta_*}{\alpha + \beta} \right) \beta^2 \|\Delta(t)\|_X^2$$
$$\leq (1 + 7\tau - \beta\eta_*(2 - \eta_*\beta)) \|\Delta(t)\|_X^2$$
$$= (q + 7\tau) \|\Delta(t)\|_X^2.$$

**Case 2**: $0 < \eta_* \leq \frac{2}{\alpha+\beta}$.

Similarly, we have

$$\|\Delta(t+1)\|_X^2 \leq (1 + 7\tau - \alpha\eta_*(2 - \eta_*\alpha)) \|\Delta(t)\|_X^2 = (q + 7\tau) \|\Delta(t)\|_X^2.$$

In both cases, we have $\|\Delta(t+1)\|_X^2 \leq (q + 7\tau) \|\Delta(t)\|_X^2$. Thus, $\|\Delta(t)\|_X^2 \leq (q + 7\tau)^t \|\Delta(0)\|_X^2$. $\square$

With the help of this lemma, we further obtain the following trajectories comparison lemma, which leads to the main conclusions of Theorem 2. We will show that the trajectories of GD (7) for deep linear neural networks (6) are close to those of GD (2) for the corresponding convex optimization problem (1). Now, we introduce the following technical lemma for trajectory comparison.

**Lemma 6** *Consider GD for deep linear neural networks (7) with learning rate $\eta < \eta_1 = \frac{2n_N}{N\beta}$ for $a_N W_{N:1}(t), t = 0, 1, \cdots$, and GD (2) with learning rate $\eta_* = \frac{N}{n_N}\eta$ for $W(t), t = 0, 1, \cdots$.*

*Assume that $C(c_1, c_2)$ exists in Lemma 3 for any $c_1, c_2 > 0$. For any $\tau \in (0, 1)$, and $\eta < \eta_1$, we can choose $c_1, c_2 > 0$ and the constant $C = C(c_1, c_2) = C'(\tau, \eta/\eta_1)$, such that inequality (16) holds, given initialization condition (14), and overparameterization condition*

$$n_{min} \geq CM^2\kappa^2 B_0 N^{2\theta} n_N. \tag{17}$$

*Then, we have*

$$\|a_N W_{N:1}(t) - W(t)\|_X^2 \leq D(\tau, q, t) \|a_N W_{N:1}(0) - W_*\|_X^2, \tag{18a}$$

$$|\mathcal{E}_{DLN}(t) - \mathcal{E}(t)| \leq \beta \left( q^{t/2}\sqrt{D(\tau, q, t)} + \frac{1}{2}D(\tau, q, t) \right) \|a_N W_{N:1}(0) - W_*\|_X^2, \tag{18b}$$

$$\mathcal{E}_{DLN}(t) \leq 3\beta(q + \tau)^t \|a_N W_{N:1}(0) - W_*\|_X^2, \tag{18c}$$

*where $D(\tau, q, t) = \min\left\{ \frac{\tau}{1-q}, 2(q+\tau)^t \right\}$, with $q$ defined in (10).*

The proof of Lemma 6 requires analyzing the dynamical system

$$\Delta(t+1) = \Delta(t) - \eta_*(\nabla L(V(t)) - \nabla L(W(t))) + R(t), t \geq 0,$$

with $\Delta(t) = V(t) - W(t)$ and $V(t) = a_N W_{N:1}(t)$. The analysis is essentially the same as the proof of Lemma 5.

## 4.4 Random Initialization Fall into the Convergence Region

In Appendices E and F, we establish that when the width of each hidden layer is sufficiently large, both Gaussian and one peak random orthogonal projections and embeddings initializations will fall into the convergence region, as defined in Definition 2. Moreover, our analysis treats the balanced initialization as a specific instance of orthogonal initialization.

To analyze the product of Gaussian random matrices, it only requires analyzing the concentration inequality (Lemma 10) for the product of the $\chi^2_{n_i}$ distribution random variables, which has already been considered in Du & Hu (2019). To analyze the product of one peak random orthogonal projections and embeddings, it only requires analyzing the concentration inequality (Lemma 16) for the product of the beta distribution, which has not been considered in Hu et al. (2020).

Given this foundation, and when combined with Lemma 3 and Lemma 6, we can corroborate Theorems 1, 2, 4, and 5.

## 4.5 Navigating Away from Bad Saddles

A critical point $x^*$ of $f$ is a bad saddle if $\lambda_{min}(\nabla^2 f(x^*)) = 0$. Kawaguchi (2016) showed that the deep linear network has bad saddles, thus in general vanishing Hessian can hinder optimization. Now, we are going to explain why do bad saddles not affect GD for overparameterized deep linear neural networks.

Theorem 2.3 in Kawaguchi (2016) showed that all bad saddles satisfy that $W_{N-1:2}$ is a non-full rank matrix. Thus, to show that trajectories of GD are away from bad saddle points, it suffices to show that $\inf_t \sigma_{min}(W_{N-1:2}(t)) > 0$. In literature, there are two main ways to avoid bad saddles for GD to train the deep linear network.

On the one hand, under the setting in Arora et al. (2018b), it has been showed that if the approximate balanced initialization satisfies $\|W_{N:1}(0) - W_*\|_F \leq \sigma_{min}(W_*) - c$, for some $0 < c < \sigma_{min}(W_*)$, then $\sigma_{min}(W_{N:1}(t)) \geq c$ through the training as well as $\|W_1(t)\| \leq (4\|W_*\|_F)^{1/N}$, and $\|W_N(t)\| \leq (4\|W_*\|_F)^{1/N}$. Thus,

$$\sigma_{min}(W_{N-1:2}(t)) \geq \frac{\sigma_{min}(W_{N:1}(t))}{\|W_1(t)\|\|W_N(t)\|} \geq \frac{c}{(4\|W_*\|)^{2/N}}.$$

On the other hand, if we assume our rescaled and overparameterized weight initialization falls into the convergence region (14), we can show that (see $\mathcal{B}(t)$ in the proof of Lemma 3)

$$\sigma_{min}(W_{N-1:2}(t)) \geq \max\left\{\frac{\sigma_{min}(W_{N:2}(t))}{\sigma_{max}(W_N(t))}, \frac{\sigma_{min}(W_{N-1:1}(t))}{\sigma_{max}(W_1(t))}\right\}.$$

Thus, $\sigma_{min}(\frac{W_{N-1:2}}{(n_{N-1:2})^{1/2}}) \geq e^{-c_1-c_2}\max\{\frac{n_1}{n_{N-1}}, \frac{n_{N-1}}{n_1}\} \geq e^{-c_1-c_2} > 0$. For the overparameterized deep linear network, GD initialized in the convergence region will force the trajectories away from all bad saddles.

**Numerical Experiments:** In Appendix G, we will discuss some empirical evidence to support the main results shown in Section 3. The figures compare the trajectories of the logarithm of loss for gradient descent in deep linear networks and the corresponding convex optimization problem. The numerical experiments demonstrate that increasing the minimal width of the hidden layers in deep linear networks leads to optimization trajectories that closely resemble those of the corresponding convex problem, irrespective of the initialization scheme. This suggests that sufficient network width stabilizes the optimization process, supporting the main theoretical results presented in the paper.

## 5 Conclusion

In this paper, we present a comprehensive convergence analysis and trajectory comparison of the gradient descent method for overparameterized deep linear neural networks with different random initializations.

A key contribution of our work is the trajectory comparison between gradient descent for deep linear neural networks and the corresponding convex optimization problem. We showed that, with appropriate initialization and sufficient width of the hidden layers, the trajectories of gradient descent for deep linear neural networks closely match those of the convex optimization problem, regardless of the depth of the network. This finding provides valuable insights into the optimization dynamics of deep linear networks and their relation to their convex counterparts.

The convergence analysis and trajectory comparison presented in this paper contribute to the growing body of work on understanding the optimization landscape and dynamics of deep neural networks. Our findings on the efficiency of gradient descent in the overparameterized regime and the role of initialization in shaping the optimization trajectories provide valuable insights that can potentially extend to more general deep neural network architectures.

There could be a debate on whether the techniques and insights for linear networks can be extended to deep non-linear networks. On the one hand, works by Ding et al. (2022) and He et al. (2020) have indicated that the loss landscapes of non-linear neural networks differ significantly from those of linear networks. Specifically, non-linear networks may contain spurious local minima, while linear networks do not. Hence, the training dynamics of non-linear networks may be quite different from those of linear networks. On the other hand, Du et al. (2019) has already demonstrated that gradient descent finds a global minimum in training for sufficiently wide deep neural networks, including multilayer fully-connected neural networks, ResNet, and convolutional ResNet. In these cases, the linear rate of convergence in terms of loss has been established for overparameterized neural networks. However, the result of the trajectory comparison has not been established.

Future research directions include exploring the generalization of our findings to nonlinear deep neural networks and investigating the impact of different optimization algorithms and regularization techniques on the convergence and trajectory properties of deep networks. Furthermore, our work motivates the development of theoretical frameworks that can provide a more comprehensive understanding of the optimization dynamics of deep neural networks.

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

# A    Proofs of Basic Properties of Semi-norm

**Lemma 7** *The loss function $L(W)$ defined in (1) satisfies the following properties: for any $W, V \in \mathbb{R}^{n_y \times n_x}$,*

1. *$L(W) = L(WP_X)$,*

2. *$\nabla L(W) = \nabla L(WP_X)P_X$,*

3. *$\langle \nabla L(W), V \rangle_F = \langle \nabla L(W), V \rangle_X$,*

4. *$\|\nabla L(W)\|_F = \|\nabla L(W)\|_X$,*

5. *$\|W\|_X \equiv \|W\|_F$ if and only if $X$ is full row rank.*

6. *If $L(W)$ is $\alpha$-strongly convex ($\alpha > 0$) with respect to the semi-norm $\|\cdot\|_X$, then $W_* P_X$ is unique, that is, if $W'$ is another global minimizer of $L(W)$, then $W_* P_X = W' P_X$.*

**Proof of Lemma 7**    The first property is a direct consequence of the definition of the projection matrix $P_X$.

Notice that

$$\frac{1}{\varepsilon}(L(W + \varepsilon \Delta W) - L(W)) = \frac{1}{\varepsilon}(L(WP_X + \varepsilon \Delta W P_X) - L(WP_X)).$$

By letting $\varepsilon \to 0$, the definition of the directional derivative implies

$$\langle \nabla L(W), \Delta W \rangle_F = \langle \nabla L(WP_X), \Delta W P_X \rangle_F = \langle \nabla L(WP_X)P_X, \Delta W \rangle_F, \forall \Delta W \in \mathbb{R}^{n_y \times n_x},$$

since $P_X = P_X^T$. This completes the proof of the second property.

The third property is obtained based on the fact that the orthogonal projection matrix satisfies $P_X = P_X^T = P_X^2$, since

$$\langle \nabla L(W), V \rangle_F = \langle \nabla L(WP_X)P_X, V \rangle_F$$
$$= \langle \nabla L(WP_X)P_X^2, VP_X \rangle_F = \langle \nabla L(WP_X)P_X, V \rangle_X = \langle \nabla L(W), V \rangle_X.$$

Set $V = \nabla L(W)$. Then the fourth property is implied by the third property.

For the fifth property, first recall that $\|W\|_X = \|WP_X\|_F$ and $P_X = X(X^T X)^\dagger X^T$. $X$ is of full row rank if and only if $P_X$ is identity matrix, which completes the proof.

To prove the last property, let $W'$ be another global minimizer of $L(W)$.

Since $L(W)$ is strongly convex, we have

$$L(W') = L(W_*) \geq L(W') + \langle \nabla L(W'), W_* - W' \rangle_X + \frac{\alpha}{2} \|W' - W_*\|_X^2. \tag{19}$$

Because $L(W_*) = L(W')$, we can show that

$$\langle \nabla L(tW_* + (1-t)W'), W_* - W' \rangle_F = 0. \tag{20}$$

Applying property 3 and setting $t = 0$ in (20), inequality (19) implies $\|W' - W_*\|_X = 0$.

By the definition of semi-norm $\|\cdot\|_X$, we obtain that

$$\|W'P_X - W_* P_X\|_F = \|W' - W_*\|_X = 0,$$

which implies that $W'P_X = W_* P_X$ for any global minimizer $W'$. Thus, $W_* P_X$ is unique.

$\square$

**Proof of Lemma 1** Because $X$ is not full row rank, we know that $I - P_X \neq 0$. There exists $W$ such that $W(I - P_X) \neq 0$. Applying the first property in Lemma 7, we have

$$L(\frac{1}{2}W + \frac{1}{2}WP_X) = L((\frac{1}{2}W + \frac{1}{2}WP_X)P_X) = L(WP_X) = \frac{1}{2}L(W) + \frac{1}{2}L(WP_X),$$

provided $W \neq WP_X$.

Hence, $L$ is not strictly convex, which implies $L$ is not strongly convex.

To prove the second property, it suffices to show that $g(W) = L(W) - \frac{\alpha(l)\lambda_{min}(XX^T)}{m}\|W\|_X^2$ is convex. It is obvious that

$$g(W) = L(W) - \frac{\alpha(l)}{m}\sum_{i=1}^{m}\|Wx_i - y_i\|_2^2 + \frac{\alpha(l)}{m}(\|WX - Y\|_F^2 - \lambda_{min}(X^TX)\|W\|_X^2). \tag{21}$$

$L(W) - \frac{\alpha(l)}{m}\sum_{i=1}^{m}\|Wx_i, y_i\|_F^2$ is convex, since $l(\cdot, y_i)$ is strongly convex. The Hessian of $\|WX - Y\|_F^2 - \lambda_{min}(W^TW)\|WP_X\|_F^2$ has no negative eigenvalue, thus the second term in (21) is also convex. This completes the proof. $\square$

**Proof of Lemma 2** Set $r = rank(X)$, and $u_1, \cdots, u_r$ be an orthonormal basis of the column space of $X$. Then, $P_X = \sum_{i=1}^{r}u_iu_i^T$.
Notice that

$$\|a_nW_{N:1}(0)\|_X^2 = \|a_nW_{N:1}(0)P_X\|_F^2 = \sum_{i=1}^{r}\|a_nW_{N:1}(0)u_i\|_2^2.$$

By assumption, we have

$$\mathbb{E}\|a_nW_{N:1}(0)\|_X^2 = \mathbb{E}\sum_{i=1}^{r}\|a_nW_{N:1}(0)u_i\|_2^2 = r.$$

The Markov inequality implies

$$\mathbb{P}(\|a_nW_{N:1}(0)\|_X^2 \geq \frac{2r}{\delta}) \leq \frac{\delta}{2}.$$

Therefore, we can bound the initial loss value as

$$L_0 - L(W_*) \leq \langle \nabla L(W_*), a_NW_{N:1}(0)X - W_* \rangle + \frac{\beta}{2}\|a_NW_{N:1}(0) - W_*\|_X^2$$

$$= \frac{\beta}{2}\|a_NW_{N:1}(0) - W_*\|_X^2$$

$$\leq \beta(\|a_NW_{N:1}(0)\|_X^2 + \|W_*\|_X^2)$$

$$\leq \beta(\frac{2r}{\delta} + \|W_*\|_X^2),$$

with probability at least $1 - \delta/2$. $\square$

## B   The Exact Statements of the Main Theorems

Other than the quantities in (11), we define

$$\mathbb{C}_4 = n_N\kappa^2 B_\delta \frac{1}{(\eta_0 - \eta)^2/\eta_0^2},$$

$$\mathbb{C}_5 = n_N\kappa^2 B_\delta \frac{C_0}{(\eta_0 - \eta)^2/\eta_0^2} + \ln N, \tag{22}$$

$$\mathbb{C}_6 = n_N\kappa^2 B_\delta \frac{C_0}{(\eta_0 - \eta)^2/\eta_0^2} + C_0\ln(\underline{N}),$$

where $\eta_0 = \frac{2n_N}{e^{2c}N\beta}$ with $c > 0$. Recall that $\eta_1 = \frac{2n_N}{N\beta}$.

**Theorem 4** *Given any $c > 0$, and $0 < \delta < 1/2$, define $\eta_0 = \frac{2n_N}{e^{2c}N\beta}$, and consider the learning rate $\eta < \eta_0$. There exists a constant $C := C(c)$, such that if*

$$n_{min} \geq C \cdot \mathbb{C}_5 \cdot N, \tag{23}$$

*then with probability at least $1 - \delta$ over the random Gaussian initialization, we have*

$$\mathcal{E}_{DLN}(t) \leq \left(1 - 4e^{-c}\frac{\frac{\eta}{\eta_0}(1 - \frac{\eta}{\eta_0})}{\kappa}\right)^t \mathcal{E}_{DLN}(0).$$

**Theorem 5** *Given any $c > 0$, and $0 < \delta < 1/2$, define $\eta_0 = \frac{2n_N}{e^{2c}\beta N}$, and consider the learning rate $\eta < \eta_0$. There exists a constant $C := C(c)$, such that if*

$$n_{min} \geq C \cdot \mathbb{C}_5, \tag{24}$$

*then with probability at least $1 - \delta$ over the random one peak projections and embeddings initialization, we have*

$$\mathcal{E}_{DLN}(t) \leq \left(1 - 4e^{-c}\frac{\frac{\eta}{\eta_0}(1 - \frac{\eta}{\eta_0})}{\kappa}\right)^t \mathcal{E}_{DLN}(0).$$

*Specially, if $n_1 = n_2 = \cdots = n_{N-1} = n \geq \min\{n_N, n_0\}$, then the requirement (24) can be replaced by*

$$n \geq C \cdot \mathbb{C}_4. \tag{25}$$

**Remark 8** *Assume $L(a_N W_N \cdots W_1) = \frac{1}{2}\|a_N W_N \cdots W_1 X - Y\|_F^2$, and $n_1 = \cdots = n_{N-1} = n$. Then for Gaussian initialization, our Theorem 4 leads to Theorem 4.1 in Du & Hu (2019). Similarly, for orthogonal initialization, our Theorem 5 leads to Theorem 4.1 in Hu et al. (2020).*

Next, we present a version of the theorem related to balanced initialization.

**Theorem 6** *Assume $n_1 = \cdots = n_{N-1} = n$. Given any $c > 0$, and $0 < \delta < 1/2$, define $\eta_0 = \frac{2n_N}{e^{2c}\beta N}$, and consider the learning rate $\eta < \eta_0$. There exists a constant $C := C(c)$, such that as long as*

$$n \geq C \cdot \mathbb{C}_4. \tag{26}$$

*then with probability at least $1 - \delta$ over special balanced initial, we have*

$$\mathcal{E}_{DLN}(t) \leq \left(1 - 4e^{-c}\frac{\frac{\eta}{\eta_0}(1 - \frac{\eta}{\eta_0})}{\kappa}\right)^t \mathcal{E}_{DLN}(0).$$

## C  Inequalities in Convex Optimization

Convex optimization has been studied for about a century. Recall the definitions and basic inequalities for $\alpha-$strongly convex and $\beta-$Lipschitz functions.

**Definition 3** *A continues differentiable function $f$ is said to be $\beta-$ Lipschitz if the gradient $\nabla f$ is $\beta-$ Lipschitz, that is if for all $x, y$,*

$$\|\nabla f(y) - \nabla f(x)\| \leq \beta \|y - x\|,$$

*$f$ is said to be $\alpha-$strongly convex if for all $x, y$, we have*

$$f(y) \geq f(x) + \langle \nabla f(x), y - x \rangle + \frac{\alpha}{2}\|y - x\|^2.$$

**Proposition 1** *If $f$ is $\alpha-$strongly convex and $\nabla f$ is $\beta-$Lipschitz with respect to a (semi-)norm, then $\alpha \leq \beta$ and*

$$\langle \nabla f(x), y - x \rangle + \frac{\alpha}{2} \|y - x\|^2 \leq f(y) - f(x) \leq \langle \nabla f(x), y - x \rangle + \frac{\beta}{2} \|y - x\|^2, \tag{27}$$

$$\langle \nabla f(x) - \nabla f(y), x - y \rangle \geq \frac{\alpha\beta}{\alpha + \beta} \|x - y\|^2 + \frac{1}{\alpha + \beta} \|\nabla f(x) - \nabla f(y)\|^2, \tag{28}$$

$$\|\nabla f(x) - \nabla f(y)\| \geq \alpha \|x - y\|, \tag{29}$$

$$f(x) - f(y) \leq \langle \nabla f(x), x - y \rangle - \frac{1}{2\beta} \|\nabla f(x) - \nabla f(y)\|^2. \tag{30}$$

**Proof of Proposition 1** We only proof the last inequality.
Let $z = y - \frac{1}{\beta}(\nabla f(y) - \nabla f(x))$. Since $f$ is convex $\beta-$Lipschitz, we have

$$f(z) - f(x) \geq \langle \nabla f(x), z - x \rangle$$

and

$$f(z) - f(y) \leq \langle \nabla f(y), z - y \rangle + \frac{\beta}{2} \|z - y\|^2.$$

Thus,

$$\begin{aligned}
f(x) - f(y) =& f(x) - f(z) + f(z) - f(y) \\
\leq& \langle \nabla f(x), x - z \rangle + \langle \nabla f(y), z - y \rangle + \frac{\beta}{2} \|z - y\|^2 \\
=& \langle \nabla f(x), x - y \rangle - \frac{1}{2\beta} \|\nabla f(x) - \nabla f(y)\|^2.
\end{aligned}$$

$\square$

Before we start to prove Lemma 3, let us first include and prove the following result.

**Lemma 8** *1. Assume $L$ is $\alpha-$strongly convex, $\alpha > 0$. Denote a global minimizer of $L$ by $W_*$. Then for any $W$,*

$$L(W_*) - L(W) \geq -\frac{1}{2\alpha} \|\nabla L(W)\|_X^2. \tag{31}$$

*2. Assume $\nabla L$ is $\beta-$Lipschitz, then*

$$L(W_*) - L(W) \leq -\frac{1}{2\beta} \|\nabla L(W)\|_X^2. \tag{32}$$

**Proof of Lemma 8** **1.** First, we know that $\nabla L(W_*) = 0$. $L$ is $\alpha-$strongly convex, which implies the inequality (27) holds. Thus

$$L(V) - L(W) \geq \langle \nabla L(W), V - W \rangle_X + \frac{\alpha}{2} \|V - W\|_X^2 =: g(V).$$

Minimizing both sides in terms of $V$ gives (31).

Now we focus on minimizing $g(V)$. Since $g(V) \in C^1$ and the global minimizer exits, we have

$$\nabla g(V^*) = \nabla L(W)P_X + \alpha(V^* - W)P_X = 0,$$

where $V^*$ is a global minimizer for $g(V)$. Thus,

$$g(V^*) = -\frac{1}{2\alpha} \|\nabla L(W)\|_X^2.$$

**2.** Applying proposition 1 to a $\beta-$Lipschitz function $\nabla L$, we obtain

$$L(W_*) - L(W)$$
$$\leq \langle \nabla L(W_*), W_* - W \rangle_X - \frac{1}{2\beta} \|\nabla L(W) - \nabla L(W_*)\|_X^2$$
$$= -\frac{1}{2\beta} \|\nabla L(W)\|_X^2.$$

□

## D   Proofs Related to Convergence Region

**Proof of Lemma 3**   To prove Lemma 3, it suffices to show that the following three properties hold $\mathcal{A}(t)$, $\mathcal{B}(t)$, and $\mathcal{C}(t)$ for all $t = 0, 1, \cdots$.

1. $\mathcal{A}(t)$:
$$L_t - L(W_*) \leq (1 - \eta\gamma)^t (L_0 - L(W_*)).$$

2. $\mathcal{B}(t)$:
$$\begin{cases} \sigma_{max}(W_{N:i+1}(t)) \leq e^{c_1}(n_{N-1:i})^{1/2}, 1 \leq i \leq N-1, \\ \sigma_{min}(W_{N:i+1}(t)) \geq e^{-c_2}(n_{N-1:i})^{1/2}, 1 \leq i \leq N-1, \\ \sigma_{max}(W_{i-1:1}(t)|_{\mathcal{R}(X)}) \leq e^{c_1}(n_{i-1:1})^{1/2}, 2 \leq i \leq N, \\ \sigma_{min}(W_{i-1:1}(t)|_{\mathcal{R}(X)}) \geq e^{-c_2}(n_{i-1:1})^{1/2}, 2 \leq i \leq N, \\ \|W_{j:i}(t)\| \leq M \cdot N^\theta(\frac{1}{n_{min}}\prod_{i-1\leq k\leq j} n_k)^{1/2}, 1 < i \leq j < N. \end{cases}$$

3. $\mathcal{C}(t)$:
$$\|W_i(t) - W_i(0)\|_F \leq \frac{2e^{2c_1}\sqrt{2\beta B}}{\sqrt{n_N}\gamma} =: R, 1 \leq i \leq N.$$

Using simultaneous induction, the proof of Lemma 3 is divided into the following 3 claims.

**Claim 1** $\mathcal{A}(0), \cdots, \mathcal{A}(t), \mathcal{B}(0), \cdots, \mathcal{B}(t) \implies \mathcal{C}(t+1)$.

**Claim 2** $\mathcal{C}(t) \implies \mathcal{B}(t)$, if $n_{min} \geq \frac{C(c_1,c_2)M^2\kappa^2 B_0}{\varepsilon^2} N^{2\theta}n_N$, where $C(c_1, c_2)$ is a positive constant only depend on $c_1, c_2$.

**Claim 3** $\mathcal{A}(t), \mathcal{B}(t) \implies \mathcal{A}(t+1)$, if $n_{min} \geq C(c_1,c_2)M^2 B_0 N^{2\theta}n_N$, where $C(c_1, c_2)$ is a positive constant only depend on $c_1, c_2$.

□

**Proof of Claim 1**   As a consequence of Lemma 8 and Lemma 7, and $\mathcal{A}(s)$, $s \leq t$, we have

$$\|\nabla L(a_N W_{N:1}(s))\|_F^2 = \|\nabla L_s - \nabla L(W_* P_X)\|_X^2$$
$$\leq 2\beta[L_s - L(W_*)] \tag{33}$$
$$\leq 2\beta(1 - \eta\gamma)^s B.$$

From $\mathcal{A}(0), \cdots, \mathcal{A}(t), \mathcal{B}(0), \cdots, \mathcal{B}(t)$, we have for any $0 \leq s \leq t$,

$$\left\|\frac{\partial L}{\partial W_i}(s)\right\|_F \leq a_N \|W_{N:i+1}(s)\| \|\nabla L(a_N W_{N:1}(s))\|_F \left\|W_{i-1:1}(s)|_{\mathcal{R}(X)}\right\|$$
$$\leq \frac{e^{2c_1}}{\sqrt{n_N}} \|\nabla L(a_N W_{N:1}(s))\|_F \tag{34}$$
$$\leq \frac{e^{2c_1}}{\sqrt{n_N}}\sqrt{2\beta(1 - \eta\gamma)^s B}.$$

Then,

$$
\begin{aligned}
\|W_i(t+1) - W_i(0)\|_F &\leq \sum_{s=0}^{t} \|W_i(s+1) - W_i(s)\|_F \\
&= \sum_{s=0}^{t} \left\| \eta \frac{\partial L}{\partial W_i}(s) \right\|_F \\
&\leq \eta \frac{e^{2c_1}}{\sqrt{n_N}} \sqrt{2\beta B} \sum_{s=0}^{t} (1 - \eta\gamma)^{s/2} \\
&\leq \eta \frac{e^{2c_1}}{\sqrt{n_N}} \sqrt{2\beta B} \sum_{s=0}^{t} (1 - \eta\gamma/2)^{s} \\
&\leq \frac{2e^{2c_1}\sqrt{2\beta B}}{\sqrt{n_N}\gamma} = R.
\end{aligned}
$$

This proves $\mathcal{C}(t+1)$. $\qquad\square$

**Proof of Claim 2** Let $\delta_i = W_i(t) - W_i(0), 1 \leq i \leq N$. Using $\mathcal{C}(t)$, we have $\|\delta_i\|_F \leq R, 1 \leq i \leq N$. Set $\varepsilon_1 = e^{-c_1/2} \min\{e^{c_1} - e^{c_1/2}, e^{-c_2/2} - e^{-c_2}, 1/2\}$.
It is suffices to show that

$$
\|W_{N:i}(t) - W_{N:i}(0)\| \leq e^{c_1/2}\varepsilon_1 (n_{N-1}n_{N-1}\cdots n_{i-1})^{1/2}, 1 < i \leq N, \tag{35}
$$

$$
\left\| (W_{i:1}(t) - W_{i:1}(0))|_{\mathcal{R}(X)} \right\| \leq e^{c_1/2}\varepsilon_1 (n_1 n_2 \cdots n_{i-1})^{1/2}, 1 \leq i < N, \tag{36}
$$

and

$$
\|W_{j:i}(t) - W_{j:i}(0)\| \leq M/2 \cdot N^\theta \left( \frac{1}{n_{min}} \prod_{i-1 \leq k \leq j} n_k \right)^{1/2}, 1 < i \leq j < N, \tag{37}
$$

because $\sigma_{min}(A+B) \geq \sigma_{min}(A) - \sigma_{max}(B) = \sigma_{min}(A) - \|B\|$ and $\sigma_{max}(A+B) \leq \sigma_{max}(A) + \sigma_{max}(B) = \|A\| + \|B\|$ (e.g. see Theorem 1.3 in Chafaı et al. (2009)).
**Case 1.** We first prove (37).
For $1 \leq i < j \leq N$, we can write $W_{j:i}(t) = (W_j(0) + \delta_j) \cdots (W_i(0) + \delta_i)$.
Expanding the above product, each term has the form:

$$
W_{j:(k_s+1)}(0) \cdot \delta_{k_s} \cdot W_{(k_s-1):(k_{s-1}+1)}(0) \cdot \delta_{k_{s-1}} \cdots \delta_{k_1} \cdot W_{(k_1-1):i}(0), \tag{38}
$$

where $i \leq k_1 < \cdots < k_s \leq j$ are positions at which perturbation terms $\delta_{k_l}$ are taken out.
Notice that the convergence region assumptions (14) implies that for any $1 < i \leq j < N$,

$$
\|W_{j:i}(0)\| \leq M/2 \cdot N^\theta \left( \prod_{i \leq k \leq j-1} n_k \cdot \max\{n_{i-1}, n_j\} \right)^{1/2} \leq M \cdot N^\theta \left( \frac{\prod_{i-1 \leq k \leq j} n_k}{n_{min}} \right)^{1/2}. \tag{39}
$$

WLOG, assume $M \geq 1$. If $i = j + 1$, then

$$
\|W_{j:i}(0)\| = \|I\| \leq M \cdot N^\theta (n_j/n_{min})^{1/2}.
$$

Assume $i > 1, j < N$, applying inequality (39) as well as the following inequality

$$
\sum_{s=1}^{j-i+1} \binom{j-i+1}{s} x^s = (1+x)^{j-i+1} - 1 \leq (1+x)^N - 1, \forall x \geq 0,
$$

we obtain that

$$\|W_{j:i}(t) - W_{j:i}(0)\|$$
$$\leq \sum_{s=1}^{j-i+1} \binom{j-i+1}{s} R^s (M \cdot N^\theta)^{s+1} n_{min}^{-s/2} (n_{i-1} \cdots n_j / n_{min})^{1/2}$$
$$\leq M \cdot N^\theta (n_{i-1} \cdots n_j / n_{min})^{1/2} [(1 + R \cdot M \cdot N^\theta / \sqrt{n_{min}})^N - 1]$$
$$\leq \varepsilon_1 M \cdot N^\theta (n_{i-1} \cdots n_j / n_{min})^{1/2}.$$

The last line holds due to the following reasons:
there exists absolute constant $A_1, A_2 > 0$ such that

$$(1+x)^N - 1 \leq A_2 x N,$$

if $x \geq 0$, $N \geq 1$, and $xN \leq A_1$. Since there exists positive constant $C(c_1, c_2)$, which only depends on $c_1, c_2$, such that when

$$n_{min} \geq \frac{C(c_1, c_2) M^2 \kappa^2 B_0}{\varepsilon^2} N^{2\theta} n_N \tag{40}$$

we can have

$$R \cdot M \cdot N^{\theta+1} / \sqrt{n_{min}} \leq A_1,$$

as well as

$$[(1 + R \cdot M \cdot N^\theta / \sqrt{n_{min}})^N - 1] \leq A_2 \cdot M \cdot R \cdot N^{\theta+1} / \sqrt{n_{min}} \leq \varepsilon_1 = \varepsilon_1(c_1, c_2).$$

**Case 2.** The proof of (35) is similar. Set $j = N$, we can save the factor $M \cdot N^\theta$ from previous calculation, which means

$$\|W_{N:i}(t) - W_{N:i}(0)\|$$
$$\leq e^{c_1/2} \sum_{s=1}^{N-i+1} \binom{N-i+1}{s} R^s (M \cdot N^\theta)^s n_{min}^{-s/2} (n_{i-1} \cdots n_{N-1})^{1/2}$$
$$\leq e^{c_1/2} (n_{i-1} \cdots n_{N-1})^{1/2} [(1 + R \cdot M \cdot N^\theta / \sqrt{n_{min}})^N - 1]$$
$$\leq e^{c_1/2} \varepsilon_1 (n_{i-1} \cdots n_{N-1})^{1/2}, i \geq 2,$$

where the last line is implied by equation (40).
**Case 3.** Similarly, we have

$$\left\| W_{j:1}(t)|_{\mathcal{R}(X)} - W_{j:1}(0)|_{\mathcal{R}(X)} \right\|$$
$$\leq e^{c_1/2} \sum_{s=1}^{j} \binom{j}{s} R^s (M \cdot N^\theta)^s n_{min}^{-s/2} (n_1 \cdots n_j)^{1/2}$$
$$\leq e^{c_1/2} (n_1 \cdots n_j)^{1/2} [(1 + R \cdot M \cdot N^\theta / \sqrt{n_{min}})^N - 1]$$
$$\leq e^{c_1/2} \varepsilon_1 (n_1 \cdots n_j)^{1/2}, j \leq N - 1$$

This proves $\mathcal{B}(t)$.

$\square$

**Proof of Claim 3** The GD (7) implies

$$W_{N:1}(t+1)$$
$$= \left(W_N(t) - \eta \frac{\partial L^N}{\partial W_N}(t)\right) \left(W_{N-1}(t) - \eta \frac{\partial L^N}{\partial W_{N-1}}(t)\right) \cdots \left(W_1(t) - \eta \frac{\partial L^N}{\partial W_1}(t)\right)$$
$$= W_{N:1}(t) - \eta \cdot a_N \sum_{i=1}^{N} W_{N:i+1}(t) W_{N:i+1}^T(t) \nabla L(a_N W_{N:1}(t)) (W_{i-1:1}(t))^T (W_{i-1:1}(t)) + E(t),$$

where $E(t)$ contains all high-order terms (those with $\eta^2$ or higher). Define a linear operator

$$P(t)[A] = a_N^2 \sum_{i=1}^{N} W_{N:i+1}(t) W_{N:i+1}^T(t) (A P_X)(W_{i-1:1}(t)|_{\mathcal{R}(X)})^T W_{i-1:1}(t)|_{\mathcal{R}(X)}, \tag{41}$$

for any $A \in \mathbb{R}^{n_N \times n_0}$.

Now we have

$$a_N W_{N:1}(t+1) = a_N W_{N:1}(t) - \eta \cdot P(t)[\nabla L(a_N W_{N:1}(t) P_X)] + a_N E(t). \tag{42}$$

Easy to check that $P(t)[\cdot]$ is a sum of positive semidefinite linear operator.

The following proposition describes the eigenvalues of the linear operator $P(t)[\cdot]$.

**Proposition 2** *Let $S_1$, $S_2$ be symmetric matrices. Suppose $S_1 = U\Lambda_1 U^T$, $S_2 = V\Lambda_2 V^T$, where $U = [u_1, u_2, \cdots, u_m]$, and $V = [v_1, v_2, \cdots, v_n]$ are othogonal matrices, and $\Lambda_1 = diag(\lambda_1, \lambda_2, \cdots, \lambda_m)$ and $\Lambda_2 = diag(\mu_1, \mu_2, \cdots, \mu_n)$ are diagonal matrices. Then the linear operator $L(A) := S_1 A S_2$ is orthogonally diagonalizable, and $L(A_{ij}) = \lambda_i \mu_j A_{ij}$, where $\lambda_i \mu_j$ represent all eigenvalues corresponding to their eigenvectors $A_{ij} = u_i v_j^T$.*

Applying this proposition and the assumption $\mathcal{B}(t)$, we obtain the upper bound and lower bound for the maximum and minimum eigenvalues of positive definite operator $P(t)$, respectively,

$$\lambda_{max}(P(t)) \le a_N^2 \sum_{i=1}^{N} \sigma_{max}^2(W_{i-1:1}(t)|_{\mathcal{R}(X)}) \cdot \sigma_{max}^2(W_{N:i+1}(t)) \le \frac{N}{n_N} e^{2c_1},$$

and

$$\lambda_{min}(P(t)) \ge a_N^2 \sum_{i=1}^{N} \sigma_{min}^2(W_{i-1:1}(t)|_{\mathcal{R}(X)}) \cdot \sigma_{min}^2(W_{N:i+1}(t)) \ge \frac{N}{n_N} e^{-2c_2}.$$

In conclusion, we have

$$\lambda_{max}(P(t)) \le \frac{N}{n_N} e^{2c_1}, \text{and } \lambda_{min}(P(t)) \ge \frac{N}{n_N} e^{-2c_2}. \tag{43}$$

With learning rate $\eta = \eta_\varepsilon = \frac{(1-\varepsilon)2n_N}{e^{6c_1+3c_2}\beta N}$, $0 < \varepsilon < 1$, we have

$$L_{t+1} - L_t$$
$$\le \langle \nabla L_t, -\eta P(t)[\nabla L_t] \rangle_X + \langle \nabla L_t, a_N E(t) \rangle_X + \frac{\beta}{2} \|\eta P(t)[\nabla L_t] - a_N E(t)\|_X^2$$
$$= \langle \nabla L_t, -\eta P(t)[\nabla L_t] \rangle + \frac{\beta}{2}\eta^2 \|P(t)[\nabla L_t]\|_X^2 + F(t)$$
$$\le -\left(\eta \lambda_{min}(P(t)) - \frac{\beta}{2}\eta^2 \lambda_{max}^2(P(t))\right) \|\nabla L_t\|_X^2 + F(t)$$
$$\le -e^{-2c_2} \frac{N}{n_N} \eta \left(1 - e^{4c_1+2c_2} \frac{\beta}{2}\eta \frac{N}{n_N}\right) \|\nabla L_t\|_X^2 + F(t),$$

where

$$F(t) = \langle \nabla L_t, a_N E(t) \rangle_X + \frac{\beta}{2} \|\eta P(t)[\nabla L_t] - a_N E(t)\|_X^2 - \frac{\beta}{2}\eta^2 \|P(t)[\nabla L_t]\|_X^2.$$

We claim that $F(t)$ is small enough, such that

$$L_{t+1} - L_t$$
$$\le -e^{-2c_2} \frac{N}{n_N} \eta \left(1 - e^{4c_1+2c_2} \frac{\beta}{2}\eta \frac{N}{n_N}\right) \|\nabla L_t\|_X^2 + F(t)$$
$$\le -e^{-3c_2} \frac{N}{n_N} \eta \left(1 - e^{6c_1+3c_2} \frac{\beta}{2}\eta \frac{N}{n_N}\right) \|\nabla L_t\|_X^2 \tag{44}$$
$$= -e^{-6(c_1+c_2)} \frac{2\varepsilon(1-\varepsilon)}{\beta} \|\nabla L_t\|_X^2.$$

Assuming this claim for the moment, we complete the proof. Combining (31) and (44), we have

$$\begin{cases} L_{t+1} - L_t \leq -e^{-6(c_1+c_2)} \frac{2\varepsilon(1-\varepsilon)}{\beta} \|\nabla L_t\|_X^2, \\ L(W_*) - L_t \geq -\frac{1}{2\alpha} \|\nabla L_t\|_X^2, \end{cases}$$

which implies

$$L_{t+1} - L(W_*) \leq \left(1 - e^{-6(c_1+c_2)} \frac{4\varepsilon(1-\varepsilon)}{\kappa}\right)(L_t - L(W_*)),$$

that is

$$L_t - L(W_*) \leq \left(1 - e^{-6(c_1+c_2)} \frac{4\varepsilon(1-\varepsilon)}{\kappa}\right)^t (L_0 - L(W_*)) = (1 - \eta\gamma)^t (L_0 - L(W_*)).$$

**Estimate** $F(t)$

Notice that

$$|F(t)|$$
$$\leq \|\nabla L_t\|_X \|a_N E(t)\|_X + \frac{\beta}{2}(2\eta\lambda_{max}(P(t)) \|\nabla L_t\|_X \|a_N E(t)\|_X + \|a_N E(t)\|_X^2)$$
$$=: I_1 + I_2.$$

From (34), we have

$$\left\|\frac{\partial L}{\partial W_i}(t)\right\|_F \leq \frac{e^{2c_1}}{\sqrt{n_N}} \|\nabla L(a_N W_{N:1}(t))\|_F = \frac{e^{2c_1}}{\sqrt{n_N}} \|\nabla L(a_N W_{N:1}(t))\|_X =: K.$$

Expanding the product

$$W_{N:1}(t+1) = \left(W_N(t) - \eta\frac{\partial L^N}{\partial W_N}(t)\right)\left(W_{N-1}(t) - \eta\frac{\partial L^N}{\partial W_{N-1}}(t)\right)\cdots\left(W_1(t) - \eta\frac{\partial L^N}{\partial W_1}(t)\right),$$

each term has the form:

$$\Delta = W_{N:(k_s+1)}(t) \cdot \eta\frac{\partial L}{\partial W_{k_s}}(t) \cdot W_{(k_s-1):(k_{s-1}+1)}(t) \cdot \eta\frac{\partial L}{\partial W_{k_{s-1}}}(t) \cdots \eta\frac{\partial L}{W_{k_1}}(t) \cdot W_{(k_1-1):1}(t),$$

where $1 \leq k_1 < k_2 < \cdots < k_s \leq N$.

As a direct consequence of inequality $\mathcal{B}(t)$ and inequality (39), we obtain

$$\|\Delta\|_X = \|\Delta P_X\|_F \leq \frac{1}{a_N\sqrt{n_N}}e^{2c_1}(\eta K)^s \left(\frac{M \cdot N^\theta}{\sqrt{n_{min}}}\right)^{s-1},$$

Recall that $E(t)$ contains all high-order terms (those with $\eta^2$ or higher) in the expansion of the product. Thus, $E(t)$ can be expressed as follows:

$$\sum_{s=2}^N \sum_{1 \leq k_1 < k_2 < \cdots < k_s \leq N} W_{N:(k_s+1)}(t) \cdot \eta\frac{\partial L}{\partial W_{k_s}}(t) \cdot W_{(k_s-1):(k_{s-1}+1)}(t) \cdot \eta\frac{\partial L}{\partial W_{k_{s-1}}}(t) \cdots \eta\frac{\partial L}{\partial W_{k_1}}(t) \cdot W_{(k_1-1):1}(t).$$

Set $\xi = \min\{(e^{-2c_2} - e^{-3c_2})/e^{4c_1+1}, \frac{1}{4}(e^{6c_1} - e^{4c_1})/e^{6c_1+1}, \frac{1}{2}(e^{6c_1} - e^{4c_1})^{1/2}/e^{4c_1+1}, 1\}$.

Recall the inequality $\binom{N}{s} \leq (eN)^s$. Thus, we have

$$
\begin{aligned}
&a_N \|E(t)\|_X \\
&\leq \frac{1}{\sqrt{n_N}} e^{2c_1} \sum_{s=2}^{N} \binom{N}{s} (\eta K)^s \left(\frac{M \cdot N^\theta}{\sqrt{n_{min}}}\right)^{s-1} \\
&\leq \frac{1}{\sqrt{n_N}} \left(\frac{M \cdot N^\theta}{\sqrt{n_{min}}}\right)^{-1} e^{2c_1} \sum_{s=2}^{N} (eN)^s (\eta K)^s \left(\frac{M \cdot N^\theta}{\sqrt{n_{min}}}\right)^s \\
&\leq \frac{1}{\sqrt{n_N}} e^{2c_1} (\eta eKN) \frac{\eta eKM \cdot N^{\theta+1}/\sqrt{n_{min}}}{1 - \eta eKM \cdot N^{\theta+1}/\sqrt{n_{min}}} \\
&\leq \xi \frac{N}{n_N} \eta \cdot e^{4c_1+1} \|\nabla L(a_N W_{N:1}(t))\|_X \ (\text{ if } \eta eKM \cdot N^{\theta+1}/\sqrt{n_{min}} < \xi/(1+\xi)) \\
&= \xi \cdot e^{4c_1+1} \left(\eta \frac{N}{n_N}\right) \|\nabla L(a_N W_{N:1}(t))\|_X .
\end{aligned}
\tag{45}
$$

Using (33) and the upper bound of $\eta$, we know that there exists constant $C(c_1, c_2)$, such that

$$n_{min} \geq C(c_1, c_2) M^2 \cdot B_0 N^{2\theta} n_N,$$

and

$$\eta eKM \cdot N^{\theta+1}/\sqrt{n_{min}} \leq \frac{2\sqrt{2}M \cdot e^{1+2c_1}\sqrt{B_0}N^\theta \sqrt{n_N}}{\sqrt{n_{min}}} = \frac{1}{C'(c_1, c_2)} \leq \frac{\xi}{2} \leq \frac{\xi}{1+\xi}.$$

Using (45), we have

$$I_1 \leq \xi \cdot e^{4c_1+1} \left(\eta \frac{N}{n_N}\right) \|\nabla L_t\|_X^2 \leq (e^{-2c_2} - e^{-3c_2}) \left(\eta \frac{N}{n_N}\right) \|\nabla L_t\|_X^2 ,$$

and

$$
\begin{aligned}
&I_2 \\
&\leq \frac{\beta}{2} \left(2\xi \cdot e^{6c_1+1} \left(\eta^2 \frac{N^2}{n_N^2}\right) \|\nabla L_t\|_X^2 + \xi^2 \cdot e^{8c_1+2} \left(\eta^2 \frac{N^2}{n_N^2}\right) \|\nabla L_t\|_X^2\right) \\
&\leq (e^{6c_1} - e^{4c_1}) \frac{\beta}{2} \eta^2 \frac{N^2}{n_N^2} \|\nabla L_t\|_X^2 .
\end{aligned}
$$

Thus, (44) valid.

This proves $\mathcal{A}(t)$.

$\square$

**Proof of Lemma 4** Due to (33), (42), (43), (45), and lemma 8, we have

$$
\begin{aligned}
\|R(t)\|_X &= \left\|a_N E(t) + \eta \left(\frac{N}{n_N}\nabla L_t - P(t)[\nabla L_t]\right)\right\|_X \\
&\leq \|a_N E(t)\|_X + \eta \max\left\{\lambda_{max}(P(t)) - \frac{N}{n_N}, \frac{N}{n_N} - \lambda_{min}(P(t))\right\} \|\nabla L_t\|_X \\
&\leq (C' \cdot \xi + \max\{e^{2c_1} - 1, 1 - e^{-2c_2}\}) \cdot \eta \frac{N}{n_N} \cdot \|\nabla L_t\|_X \\
&\leq \frac{2\sqrt{2\beta(L_t - L(W_*))}}{e^{6c_1+3c_2} \cdot \beta} \cdot (C' \cdot \xi + \max\{e^{2c_1} - 1, 1 - e^{-2c_2}\}).
\end{aligned}
$$

Due to the fact that $L_t - L(W_*)$ is non-increasing in $t$, and $C'$ is a constant only depend on $c_1, c_2$, we can choose small enough positive $c_1, c_2$ and $\xi$, which depends on $\tau$, such that

$$\|R(t)\|_X \le \tau \frac{\sqrt{2\beta(L_t - L(W_*))}}{\beta} \le \tau \|a_N W_{N:1}(t) - W_*\|_X.$$

$\square$

**Proof of Lemma 6**   Using Lemma 4, we obtain that for any $\tau \in (0,1)$ and $\eta < \eta_1$, we can find small enough positive constant $c_1, c_2$, which are only depend on $\tau, \eta/\eta_1$, and constant $C = C(c_1, c_2) = C''(\tau, \eta/\eta_1)$ mentioned in Lemma 4, such that

$$\eta = \frac{(1-\varepsilon)2n_N}{e^{6c_1+3c_2}\beta N},$$

where $0 < \varepsilon < 1$, as well as

$$V(t+1) = V(t) - \eta_*\nabla L(V(t)) + R(t),$$

where $V(t) = a_N W_{N:1}(t)$, $\eta_* = \frac{N}{n_N}\eta$, and $\|R(t)\|_X \le \tau' = \tau \|V(t) - W_*\|_X$.

Notice that $\theta_0 := \eta/\eta_1 = \frac{1-\varepsilon}{e^{6c_1+3c_2}}$ and $\eta/\eta_0 = 1 - \varepsilon$, where $\eta_0 = \frac{2n_N}{e^{6c_1+3c_2}\beta N}$.

For the right hand side of inequality (15), we have

$$\frac{C(c_1, c_2)M^2\kappa^2 B_0}{\varepsilon^2}N^{2\theta}n_N = \frac{C''(\tau, \eta/\eta_1)M^2\kappa^2 B_0}{\varepsilon^2}N^{2\theta}n_N.$$

To show that inequality (15) is equivalent to inequality (17), it suffices to show that $\varepsilon$ only depend on $\tau, \eta/\eta_1$. Notice that

$$\varepsilon = 1 - \eta/\eta_0 = 1 - \theta_0 e^{6c_1+3c_2},$$

and $c_1, c_2$ only depend on $\tau$ and $\eta/\eta_1$, which implies $\varepsilon$ only depend on $\tau, \eta/\eta_1$.

Now, we start to prove the three inequalities in (18).

Recall GD (2) for $W(t)$. Define $\Delta(t) = V(t) - W(t) = a_N W_{N:1}(t) - W(t)$. Notice that

$$\Delta(t+1) = \Delta(t) - \eta_*(\nabla L(V(t)) - \nabla L(W(t))) + R(t),$$

and

$$\begin{aligned}
&\|\Delta(t+1)\|_X^2 \\
\le& \eta_*^2\|\nabla L(V(t)) - \nabla L(W(t))\|_X^2 - 2\eta_*\langle\Delta(t), \nabla L(V(t)) - \nabla L(W(t))\rangle_X \\
& + \|\Delta(t)\|_X^2 + (2\|\Delta(t)\|_X + 2\eta_*\|\nabla L(V(t)) - \nabla L(W(t))\|_X + \tau')\tau'.
\end{aligned}$$

Let $l_t = 2\|\Delta(t)\|_X + 2\eta_*\|\nabla L(V(t)) - \nabla L(W(t))\|_X + \tau'$.

Now, we aim to find an upper bound for $l_t$.

Applying lemma 8 with the assumption $0 < \eta_* = \frac{N}{n_N}\eta < \frac{2}{\beta}$, we know that

$$l_t \le (6\|\Delta(t)\|_X + \tau') \le 7(\|W(t) - W_*\|_X + \|V(t) - W_*\|_X). \tag{46}$$

Thus

$$l_t\tau' \le 7\tau \|V(t) - W_*\|_X (\|V(t) - W_*\|_X + \|W(t) - W_*\|_X) =: U_t\tau.$$

By inequality (28),

$$
\begin{aligned}
&\|\Delta(t+1)\|_X^2 \\
&\leq \|\Delta(t)\|_X^2 - 2\eta_* \langle \Delta(t), \nabla L(V(t)) - \nabla L(W(t)) \rangle_X \\
&\quad + \eta_*^2 \|\nabla L(V(t)) - \nabla L(W(t))\|_X^2 + U_t \tau \\
&= \|\Delta(t)\|_X^2 - 2\eta_* \langle V(t) - W(t), \nabla L(V(t)) - \nabla L(W(t)) \rangle_X \\
&\quad + \eta_*^2 \|\nabla L(V(t)) - \nabla L(W(t))\|_X^2 + U_t \tau \\
&\leq \|\Delta(t)\|_X^2 - 2\eta_* \frac{\alpha\beta}{\alpha+\beta} \|\Delta(t)\|_X^2 \\
&\quad + \left( \eta_*^2 - \frac{2\eta_*}{\alpha+\beta} \right) \|\nabla L(V(t)) - \nabla L(W(t))\|_X^2 + U_t \tau.
\end{aligned}
$$

**Case 1**: $\frac{2}{\alpha+\beta} < \eta_* < \frac{2}{\beta}$.
In this case, we have

$$
\begin{aligned}
&\|\Delta(t+1)\|_X^2 \\
&\leq \|\Delta(t)\|_X^2 - 2\eta_* \frac{\alpha\beta}{\alpha+\beta} \|\Delta(t)\|_X^2 + \left( \eta_*^2 - \frac{2\eta_*}{\alpha+\beta} \right) \|\nabla L(V(t)) - \nabla L(W(t))\|_X^2 + U_t \tau \\
&\leq \|\Delta(t)\|_X^2 - 2\eta_* \frac{\alpha\beta}{\alpha+\beta} \|\Delta(t)\|_X^2 + \left( \eta_*^2 - \frac{2\eta_*}{\alpha+\beta} \right) \beta^2 \|\Delta(t)\|_X^2 + U_t \tau \\
&\leq (1 - \beta\eta_*(2 - \eta_*\beta)) \|\Delta(t)\|_X^2 + U_t \tau \\
&=: q \|\Delta(t)\|_X^2 + U_t \tau.
\end{aligned}
$$

**Case 2**: $0 < \eta_* \leq \frac{2}{\alpha+\beta}$.
Similarly, we have

$$
\|\Delta(t+1)\|_X^2 \leq (1 - \alpha\eta_*(2 - \eta_*\alpha)) \|\Delta(t)\|_X^2 + U_t \tau =: q \|\Delta(t)\|_X^2 + U_t \tau. \tag{47}
$$

In both cases, we have $0 < q < 1$.

First of all, since $U_t \leq U_0$ and $\|\Delta(0)\|_X = 0$, we obtain that

$$
\|\Delta(t)\|_X^2 \leq \frac{U_0 \tau}{1-q} + q^t \left( \|\Delta(0)\|_X^2 - \frac{U_0 \tau}{1-q} \right) \leq \frac{U_0 \tau}{1-q} \leq \frac{14\tau}{1-q} \|V(0) - W_*\|_X^2.
$$

Applying Lemma 5 for $V(t)$ and $W(t)$, we obtain $\|V(t) - W_*\|_X^2 \leq (1+\varepsilon)^t q^t \|V(0) - W_*\|_X^2$ and $\|W(t) - W_*\|_X^2 \leq q^t \|W(0) - W_*\|_X^2$, respectively. Thus,

$$
\begin{aligned}
&|L(W(t)) - L(a_N W_{N:1}(t))| \\
&\leq |\langle \nabla L(W(t)), \Delta(t) \rangle_X| + \frac{\beta}{2} \|\Delta(t)\|_X^2 \\
&\leq \beta \|W(t) - W_*\|_X \cdot \|\Delta(t)\|_X + \frac{\beta}{2} \|\Delta(t)\|_X^2 \\
&\leq \beta \left( q^{t/2} \sqrt{\frac{14\tau}{1-q}} + \frac{7\tau}{1-q} \right) \|V(0) - W_*\|_X^2.
\end{aligned}
$$

Generally speaking, (47) implies

$$
\|\Delta(t)\|_X^2 \leq \tau \sum_{j=0}^{t-1} q^{t-1-j} U_j.
$$

We have

$$\|\Delta(t)\|_X^2 \leq 14\tau \sum_{j=0}^{t-1} (q+7\tau)^j q^{t-1-j} \|V(0) - W_*\|_X^2$$

$$\leq 2(q+7\tau)^t \left(1 - (\frac{q}{q+7\tau})^t\right) \|V(0) - W_*\|_X^2$$

Thus, we have

$$\|a_N W_{N:1}(t) - W(t)\|_X^2 \leq \min\left\{\frac{14\tau}{1-q}, 2(q+7\tau)^t\right\} \|V(0) - W_*\|_X^2,$$

as well as

$$|L(W(t)) - L(a_N W_{N:1}(t))|$$

$$\leq \beta \|W(t) - W_*\|_X \cdot \|\Delta(t)\|_X + \frac{\beta}{2} \|\Delta(t)\|_X^2$$

$$\leq \beta \left(\sqrt{\min\left\{\frac{14\tau}{1-q}, 2(q+7\tau)^t\right\}} \cdot q^{t/2} + \frac{1}{2}\min\left\{\frac{14\tau}{1-q}, 2(q+7\tau)^t\right\}\right) \|V(0) - W_*\|_X^2.$$

By triangle inequality as well as $L(W(t)) - L(W_*) \leq \frac{\beta}{2} q^t \|V(0) - W_*\|_X^2$, we have

$$|L(a_N W_{N:1}(t)) - L(W_*)| \leq 3\beta(q+7\tau)^t \|V(0) - W_*\|_X^2.$$

Without loss of generality, we replace all $14\tau$ and $7\tau$ by $\tau$, which completes the proof. $\qquad\square$

## E   Gaussian Initialization Fall into the Convergence Region

In this section, we first establish some spectral properties of the products of random Gaussian matrices. The spectral properties lead to the conclusion that overparameterization guarantees that the random initialization will fall into the convergence region with high probability. Denote by $N(0,1)$ the standard Gaussian distribution, and $\chi_k^2$ the chi square distribution with $k$ degrees of freedom. Let $S^{d-1} = \{x \in \mathbb{R}^d; \|x\|_2 = 1\}$ be the unit sphere in $\mathbb{R}^d$.
The scaling factor $a_N = \frac{1}{\sqrt{n_1 n_2 \cdots n_N}}$ ensures that the networks at initialization preserves the norm of every input in expectation.

**Lemma 9** *For any $x \in \mathbb{R}^{n_0}$, the Gaussian initialization satisfies*

$$\mathbb{E}\left[\|a_N W_{N:1}(0)x\|_2^2\right] = \|x\|_2^2.$$

**Proof of Lemma 9**   For random matrix $A \in \mathbb{R}^{n_i \times n_{i-1}}$ with i.i.d $N(0,1)$ entries and any vector $0 \neq v \in \mathbb{R}^{n_{i-1}}$, the distribution of $\frac{\|Av\|_2^2}{\|v\|_2^2}$ is $\chi_{n_i}^2$. We rewrite

$$\|W_{N:1}(0)x\|_2^2 / \|x\|_2^2 = Z_N Z_{N-1} \cdots Z_1,$$

where $Z_i = \|W_{i:1}(0)x\|^2 / \|W_{i-1:1}(0)x\|^2$.
Then we know that the distribution of random variable $Z_1 \sim \chi_{n_1}^2$, and conditional distribution of random variables $Z_i | (Z_1, \cdots, Z_{i-1}) \sim \chi_{n_i}^2 (1 < i \leq N)$. Thus, $Z_1, \cdots, Z_{n_i}$ are independent. By law of iterated expectations, we have

$$\mathbb{E}[\|W_{N:1}(0)x\|_2^2 / \|x\|_2^2] = \prod_{j=1}^{N} n_j.$$

$\qquad\square$

Define $\Delta_1 = \sum_{j=1}^{N-1} 1/n_j$. Now, we introduce a new notation $\Omega\left(\frac{1}{\Delta_1}\right)$, which means that there exists $k > 0$, such that $\Omega\left(\frac{1}{\Delta_1}\right) \geq \frac{k}{\Delta_1}$.

**Lemma 10** *Consider real random matrix $A_j \in \mathbb{R}^{n_j \times n_{j-1}}, 1 \leq j \leq q$ with i.i.d $N(0,1)$ entries and any vector $0 \neq x \in \mathbb{R}^{n_1}$.*
*Define $\Delta_1(q) = \sum_{j=1}^{q} \frac{1}{n_j}$ and $n_{min} = \min_{1 \leq j \leq q} n_j$. Then*

$$\mathbb{P}(\|A_q A_{q-1} \cdots A_1 x\|_2^2 / \|x\|_2^2 > e^c n_1 \cdots n_q) \leq \exp\left\{-\frac{c^2}{8\Delta_1(q)}\right\} =: f_1(c), \forall c > 0. \tag{48}$$

*When $0 < c \leq 3\ln 2$, $\Delta_1(q) \leq c/(12\ln 2)$, we have*

$$\mathbb{P}(\|A_q A_{q-1} \cdots A_1 x\|_2^2 / \|x\|_2^2 < e^{-c} n_1 \cdots n_q) \leq \exp\left\{-\frac{c^2}{36\ln(2)\Delta_1(q)}\right\} =: f_2(c). \tag{49}$$

*Hence, for any $x \in S^{n_0 - 1}$ with probability at least $1 - e^{-\Omega\left(\frac{1}{\Delta_1(q)}\right)}$, we have*
$$e^{-c_2/2}(n_1 \cdots n_q)^{1/2} \leq \|A_q \cdots A_1 x\|_2 \leq e^{c_1/2}(n_1 \cdots n_q)^{1/2},$$

*when $0 < c_2 \leq 3\ln 2$, $\Delta_1(q) \leq c_2/(12\ln 2)$.*

**Proof of Lemma 10** For random matrix $A_i \in \mathbb{R}^{n_i \times n_{i-1}}$ with i.i.d $N(0,1)$ entries and any vector $0 \neq v \in \mathbb{R}^{n_{i-1}}$, the random variable $\frac{\|A_i v\|_2^2}{\|v\|_2^2}$ is distributed as $\chi_{n_i}^2$. We rewrite

$$\|A_q \cdots A_1 x\|_2^2 / \|x\|_2^2 = Z_q Z_{q-1} \cdots Z_1,$$

where $Z_i = \|A_{i:1} x\|^2 / \|A_{i-1:1} x\|^2$. We have $Z_1 \sim \chi_{n_1}^2$, $Z_i | (Z_1, \cdots, Z_{i-1}) \sim \chi_{n_i}^2 (1 < i \leq q)$.
Recall the moments of $Z \sim \chi_m^2$:
$$\mathbb{E}[Z^\lambda] = \frac{2^\lambda \Gamma(\frac{m}{2} + \lambda)}{\Gamma(\frac{m}{2})}, \forall \lambda > -\frac{m}{2}.$$

Now, we aim to find the Chernoff type bound.

**Case 1:** We define ratio of Gamma function

$$R(x, \lambda) = \frac{\Gamma(x + \lambda)}{\Gamma(x)}, \lambda > 0, x > 0.$$

In Jameson (2013), we have

$$R(x, \lambda) \leq x(x + \lambda)^{\lambda - 1} \leq (x + \lambda)^\lambda, \lambda > 0, x > 0. \tag{50}$$

Fixed $c > 0$, for any $\lambda > 0$ we have

$$
\begin{aligned}
\mathbb{P}(Z_q \cdots Z_1 > e^c n_1 \cdots n_q) &\leq \mathbb{P}((Z_q \cdots Z_1)^\lambda > e^{\lambda c}(n_1 \cdots n_q)^\lambda) \\
&\leq e^{-\lambda c}(n_1 \cdots n_q)^{-\lambda} \mathbb{E}[(Z_q \cdots Z_1)^\lambda] &&\text{(Markov inequality)} \\
&= \exp\{-\lambda(c + \ln(n_1 \cdots n_q))\} \prod_{j=1}^{q} 2^\lambda R(n_j/2, \lambda) &&\text{(Law of total expectation)} \\
&\leq \exp\{-\lambda(c + \ln(n_1 \cdots n_q)) + q\lambda \ln 2 + \sum_{j=1}^{q} \lambda \ln(\frac{n_j}{2} + \lambda)\} &&\text{(Inequality (50))} \\
&= \exp\{-\lambda c + \lambda \sum_{j=1}^{q} \ln(1 + \frac{2\lambda}{n_j})\} \\
&\leq \exp\{-\lambda c + 2\lambda^2 \sum_{j=1}^{q} \frac{1}{n_j}\}.
\end{aligned}
$$

Define constant $\Delta_1(q) = \sum_{j=1}^{q} \frac{1}{n_j}$. Set $\lambda = \frac{c}{4\Delta_1(q)}$, we obtain (48).

**Case 2:** Let $n_{min} = \min_{1 \leq j \leq q} n_j$.

$$\mathbb{P}(Z_q \cdots Z_1 < e^{-c} n_1 \cdots n_q) \leq \mathbb{P}((Z_q \cdots Z_1)^{\lambda} > e^{-\lambda c}(n_1 \cdots n_q)^{\lambda})$$

$$\leq \exp\{\lambda(c - \ln(n_1 \cdots n_q)) + q\lambda \ln 2 + \sum_{j=1}^{q} \ln R(\frac{n_j}{2}, \lambda)\}.$$

Define

$$f(\lambda) = \lambda(c - \ln(n_1 \cdots n_q)) + q\lambda \ln 2 + \sum_{j=1}^{q} \ln R(\frac{n_j}{2}, \lambda), -\frac{n_{min}}{2} < \lambda \leq 0.$$

Notice that $f(0) = 0$. Define digamma function,

$$\psi(x) = \frac{d}{dx} \ln(\Gamma(x)) = \frac{\Gamma'(x)}{\Gamma(x)}.$$

Qi et al. (2006) proved the following sharp inequality of digamma function,

$$\ln(x + \frac{1}{2}) - \frac{1}{x} < \psi(x) < \ln(x + e^{-\gamma}) - \frac{1}{x}, x > 0,$$

where $\gamma$ is the Euler-Mascheroni constant, and $e^{-\gamma} \approx 0.561459$.
Thus,

$$f'(\lambda) = c + \sum_{j=1}^{q} \left[ -\ln(\frac{n_j}{2}) + \psi(\frac{n_j}{2} + \lambda) \right] \geq c + \sum_{j=1}^{q} \ln(1 + \frac{\lambda + 1/2}{n_j/2}) - \sum_{j=1}^{q} \frac{1}{n_j/2 + \lambda}.$$

Since $\ln(1 + x)$ is concave, we have

$$\ln(1 + x) \geq 2\ln(2)x, x \in [-1/2, 0].$$

If $-\frac{n_{min}}{4} \leq \lambda \leq 0$, then

$$f(\lambda) = f(0) - \int_{\lambda}^{0} f'(x)dx$$

$$\leq c\lambda + \int_{0}^{\lambda} \left[ \sum_{j=1}^{q} \ln(1 + \frac{x + 1/2}{n_j/2}) - \sum_{j=1}^{q} \frac{1}{n_j/2 + x} \right] dx$$

$$= c\lambda + \sum_{j=1}^{q} \left[ \lambda \ln(1 + \frac{\lambda + 1/2}{n_j/2}) + (n_j/2 + 1/2)\ln(1 + \frac{\lambda}{n_j/2 + 1/2}) - \lambda - \ln(1 + \frac{\lambda}{n_j/2}) \right]$$

$$\leq c\lambda + \sum_{j=1}^{q} (\lambda - 1)\ln(1 + \frac{\lambda}{n_j/2})$$

$$\leq c\lambda + 4\ln(2)\lambda(\lambda - 1)\Delta_1(q).$$

Assume $0 < c \leq 3\ln 2$. Let $A = 12\ln 2$, and $\lambda^* = -\frac{c}{A\Delta_1(q)}$. Since $n_{min}\Delta_1(q) \geq 1$, we have $\lambda^* \geq -n_{min}/4$. Assume $\Delta_1(q) \leq c/(12\ln 2)$.
Thus

$$f(\lambda^*) \leq -\frac{c^2}{A\Delta_1(q)} + 4\ln 2 \frac{c^2}{A\Delta_1(q)} \left( \frac{\Delta_1(q)}{c} + \frac{1}{A} \right) \leq -\frac{c^2}{36\ln(2)\Delta_1(q)}.$$

Thus, we obtain (49). $\qquad\square$

**Lemma 11** *There exists a positive constant $C(c_1, c_2)$ which only depends on $c_1, c_2$, such that if $n_N \Delta_1 \leq C(c_1, c_2)$, then for any fixed $1 < i \leq N$, with probability at least $1 - \exp\left\{ -\Omega\left(\frac{1}{\Delta_1}\right) \right\}$ we have*

$$\sigma_{\max}(W_{N:i}(0)) \leq e^{c_1}(n_{i-1}n_i \cdots n_{N-1})^{1/2},$$

*and*

$$\sigma_{\min}(W_{N:i}(0)) \geq e^{-c_2}(n_{i-1}n_i \cdots n_{N-1})^{1/2}.$$

**Proof of Lemma 11** Let $A = W_{N:i}^T(0)$. We know that

$$\sigma_{max}(A) = \|A\| = \sup_{v \in S^{n_N-1}} \|Av\|_2$$

and

$$\sigma_{min}(A) = \inf_{v \in S^{n_N-1}} \|Av\|_2 \,.$$

Applying lemma 10, we know that with probability at least $1 - \exp\left\{-\Omega\left(\frac{1}{\Delta_1}\right)\right\}$,

$$\|Av\|_2 / \|v\|_2 \in [e^{-c_2/2}P, e^{c_1/2}P],$$

where $P = (n_{i-1} \cdots n_{N-1})^{1/2}$.
Set $\phi = \min\{1 - e^{-c_1/2}, (e^{-c_2/2} - e^{-c_2})/(e^{-c_2/2} + e^{c_1})\}$. Take a $\phi$-net $\mathcal{N}_\phi$ for $S^{n_N-1}$ with size $|\mathcal{N}_\phi| \le (3/\phi)^{n_N}$.
Notice that with this size we can actually cover the unit ball, not only the unit sphere.
Thus, with probability at least $1 - |\mathcal{N}_\phi| \exp\left\{-\Omega\left(\frac{1}{\Delta_1}\right)\right\}$, for all $u \in \mathcal{N}_\phi$ simultaneously we have

$$\|Au\|_2 / \|u\|_2 \in [e^{-c_2/2}P, e^{c_1/2}P].$$

Fixed $v \in S^{n_N-1}$, there exists $u \in \mathcal{N}_\phi$ such that $\|u - v\|_2 \le \phi$. WLOG, we assume $1 - \phi \le \|u\|_2 \le 1$. We obtain

$$\|Av\|_2 \le \|Au\|_2 + \|A(u-v)\|_2 \le e^{c_1/2}P + \phi \|A\| \,.$$

Taking supereme over $\|v\|_2 = 1$, we obtain

$$\sigma_{max}(A) = \|A\| \le \frac{e^{c_1/2}}{1-\phi}P \le e^{c_1}P.$$

For the lower bound, we have

$$\|Av\|_2 \ge \|Au\|_2 - \|A(u-v)\|_2 \ge e^{-c_2/2}P \|u\| - \phi \|A\| \ge \left[(1-\phi)e^{-c_2/2} - \phi e^{c_1}\right]P \ge e^{-c_2}P.$$

Taking the infimum over $\|v\|_2 = 1$, we get

$$\sigma_{min}(A) \ge e^{-c_2}P.$$

The conclusions hold with probability at least

$$1 - |\mathcal{N}_\phi| \exp\left\{-\Omega\left(\frac{1}{\Delta_1}\right)\right\}$$
$$\ge 1 - \exp\{n_N \ln(3/\phi)\} \exp\left\{-\Omega\left(\frac{1}{\Delta_1}\right)\right\}$$
$$\ge 1 - \exp\left\{-\Omega\left(\frac{1}{\Delta_1}\right)\right\},$$

since $n_N \Delta_1 \le C(c_1, c_2)$. $\qquad\square$

**Lemma 12** *There exists a positive constant $C(c_1, c_2)$ which only depends on $c_1, c_2$, such that if $rank(X)\Delta_1 \le C(c_1, c_2)$, then for any fixed $1 \le j < N$, with probability at least $1 - \exp\{-\Omega\left(\frac{1}{\Delta_1}\right)\}$ we have*

$$\sigma_{\max}(W_{j:1}(0)|_{\mathcal{R}(X)}) \le e^{c_1}(n_1 n_2 \cdots n_j)^{1/2},$$

*and*

$$\sigma_{\min}(W_{j:1}(0)|_{\mathcal{R}(X)}) \ge e^{-c_2}(n_1 n_2 \cdots n_j)^{1/2}.$$

**Proof of Lemma 12**    The proof is similar to that of previous lemma. The only difference is that now we consider the $\phi-$net to cover the unit sphere in $\mathcal{R}(X) \cap \mathbb{R}^{n_0}$, with $\dim \mathcal{R}(X) \cap \mathbb{R}^{n_0} = rank(X)$, where $\mathcal{R}(X)$ represents the column space of $X$. $\hfill\square$

**Lemma 13**  *Set $C = n_{max}/n_{min} < \infty$, $\theta = 1/2$. Assume $\Omega(1/\Delta_1) \geq \frac{k}{\Delta_1}$, where $0 < k < 1$ is a constant and $\Delta_1$ satisfies*

$$
\begin{cases}
\Delta_1 \leq \min\left\{ \frac{k}{5\ln(6)}, \frac{k}{5\ln(5\ln(6)e/k)} \right\} \\
\Delta_1 \ln(C) \leq \min\left\{ \frac{k}{5\ln(5\ln(6)e/k)}, \frac{k}{5} \right\} \\
\Delta_1 \ln(N^{2\theta}) \leq k/5.
\end{cases}
$$

*Given $1 < i \leq j < N$, with probability at least $1 - 2e^{-k/(5\Delta_1)} = 1 - e^{-\Omega(1/\Delta_1)}$ we have*

$$
\|W_{j:i}(0)\| \leq M_k \sqrt{C} N^\theta (n_i \cdots n_{j-1} \cdot \max\{n_{i-1}, n_j\})^{1/2},
$$

*where $M_k$ is a positive constant that only depends on $k$.*

**Proof of Lemma 13**    WLOG, assume $n_{i-1} \leq n_j$. Let $A = W_{j:i}(0)$. From lemma 10, we know that fixed $v \in S^{n_{i-1}-1}$, with probability at least $1 - e^{-\Omega(1/\Delta_1)}$ we have $\|Av\|_2 \leq 4/3(n_i \cdots n_j)^{1/2}$. .

Take a small constant $c = \frac{kN^{2\theta}}{5\ln(6)\Delta_1 n_{i-1}} \geq \frac{k}{5\ln(6)C}$. Let $v_1, \cdots, v_{n_{i-1}}$ be an orthonormal basis for $R^{n_{i-1}}$. Partition the index set $\{1, 2, \cdots, v_{n_{i-1}}\} = S_1 \cup S_2 \cup \cdots \cup S_{\lceil N^{2\theta}/c \rceil}$, where $|S_l| \leq \lceil cn_{i-1}/N^{2\theta} \rceil$ for each $1 \leq l \leq \lceil N^{2\theta}/c \rceil$.

The following discussion is similar to the proof of lemma 11, hence we omit some details. For each $l$, taking a $1/2-$ net $\mathcal{N}_l$ for the set $V_{S_l} = \{v \in S^{n_{i-1}-1}; v \in span\{v_i; i \in S_l\}\}$, we can get

$$
\|Au\|_2 \leq 4(n_i \cdots n_j)^{1/2}, u \in V_{S_l},
$$

with probability at least

$$
1 - |\mathcal{N}_l|e^{-k/\Delta_1} \geq 1 - \exp\{-k/\Delta_1 + (cn_{i-1}/N + 1)\ln 6\} \geq 1 - e^{-3k/(5\Delta_1)},
$$

since $\Delta_1 \leq \frac{k}{5\ln(6)}$.

Therefore, for any $v \in \mathbb{R}^{n_{i-1}}$, we can write it as the sum $v = \sum_l a_l v_l$, where $\alpha_l \in \mathbb{R}$ and $v_l \in V_{S_l}$ for each $l$. We also know that $\|v\|_2^2 = \sum_{l \geq 1} |\alpha_l|^2$.

Then we have

$$
\|Av\|_2 \leq \sum_l |\alpha_l| \|Av_l\|_2 \leq 4(n_i \cdots n_j)^{1/2} \sqrt{\lceil N^{2\theta}/c \rceil \sum_l |a_l|^2} \leq M_k \sqrt{C} N^\theta (n_i \cdots n_j)^{1/2} \|v\|_2.
$$

Thus,

$$
\|A\| \leq M_k \sqrt{C} N^\theta (n_i \cdots n_j)^{1/2}.
$$

Notice that when $C \leq e$, $\Delta_1 \leq \frac{k}{5\ln(5\ln(6)e/k)} \leq \frac{k}{5\ln(5\ln(6)\cdot C/k)}$, and when $C > e$, we have

$$
\Delta_1 \ln(C) \leq \min\left\{ \frac{k}{5\ln(5\ln(6)e/k)}, k/5 \right\} \leq \frac{k\ln(C)}{5\ln(5\ln(6)\cdot C/k)}.
$$

The success probability is at least

$$
1 - \lceil N^{2\theta}/c \rceil \cdot e^{-3k/(5\Delta_1)}
$$
$$
\geq 1 - \exp\left\{ \ln\left(\frac{5\ln(6)\cdot C}{k}\right) + \ln(N^{2\theta}) - 3k/(5\Delta_1) \right\} - e^{-3k/(5\Delta_1)}
$$
$$
\geq 1 - 2e^{-k/(5\Delta_1)},
$$

since

$$
\Delta_1 \leq \frac{k}{5\ln(5\ln(6)\cdot C/k)} \text{ and } \Delta_1 \ln(N^{2\theta}) \leq k/5.
$$

$\hfill\square$

**Proof of Theorem 4**   The requirement on size $\{n_1, n_2, \cdots, n_{N-1}, N\}$ in (23) makes sure that lemma 11, 12, 13, 2, and 3 hold.

WLOG, we set $c_1 = c/6, c_2 = c/3$, $M = 2M_k\sqrt{C_0}$, $B_0 = B_\delta$, and $\eta =: \frac{(1-\varepsilon)2n_N}{e^{2c}\beta N}$, then with probability at least

$$1 - N^2 e^{-\Omega(1/\Delta_1)} - \delta/2 \geq 1 - \delta, \text{ since } \Delta_1 \leq \frac{1}{C(c)} \min\left\{\frac{1}{\ln N}, \frac{1}{\ln(1/\delta)}\right\},$$

the random initialization satisfies the initialization assumption (14) and the overparameterization assumption (15). Applying Lemma 3, we complete the proof. $\qquad\square$

# F   Orthogonal Initialization Fall into the Convergence Region

There are some basic facts for random projections and embeddings. Most of the following properties can be found in Eaton (1989).

**Proposition 3**

1. $A$ is a random embedding if and only if $A^T$ is a random projection.

2. If $A$ is a square matrix, then random projection, random embedding and random orthogonal matrix are equivalent.

3. The uniform distribution on the group is a left and right invariant probability measure, that is, if $A$ is a random orthogonal matrix, then $A, UA, AU$ are all random orthogonal matrix, where $U$ is a non-random orthogonal matrix.

4. Assume $X$ is a $n \times q(q \leq n)$ random matrix whose entries are i.i.d. $N(0,1)$ random variables. Then $A := X(X^T X)^{-1/2}$ is a random embedding, since $A^T A = I_q$ and the distribution of $A$ is left invariant, which means that $A$ and $UA$ have the same distribution, where $U$ is a non-random orthogonal matrix.

5. If $A$ is a uniform distribution over an orthogonal group of order $n$ and $A$ is partitioned as $A = (A_1, A_2)$, where $A_1$ is $n \times q$ and $A_2$ is $n \times (n - q)$, then $A_1^T$ and $A_2^T$ are both random orthogonal matrix.

6. The columns of uniform distribution over orthogonal group of order $n$, and

$$\frac{(\xi_1, \cdots, \xi_n)}{\sqrt{\xi_1^2 + \xi_2^2 + \cdots + \xi_n^2}}$$

have the same distribution, where $\xi_1, \cdots, \xi_n$ are i.i.d. $N(0,1)$ random variables.

7. Assume $A = A_{n \times p}, n \leq p$ is a random orthogonal projection. For any $v \in S^{p-1}$, $\|Av\|_2^2$ and $(\sum_{i=1}^n \xi_i^2)/(\sum_{j=1}^p \xi_j^2)$ are both following beta distribution with $\alpha = n/2, \beta = (p-n)/2$, where $\xi_1, \cdots, \xi_n$ are i.i.d. $N(0,1)$ random variables.

**Remark 9**   There are several ways to construct random matrix $A = (a_{ij})_{q \times n}$, $q \leq n$, which is uniformly distributed over rectangular matrices with $AA^T = c^2 I_q, c > 0$. Let $O_n$ be uniformly distributed over real orthogonal group of order $n$, and $O_n$ is partitioned as $O_n = (A_1^T, A_2^T)^T$, where $A_1$ is $q \times n$. Assume $X = (x_{ij})_{q \times n}$, and $x_{ij}$ are independent standard normal random variables. Then $A, cA_1$, and $c(XX^T)^{-1/2}X$ have the same distribution.

**Lemma 14**   For any $x \in \mathbb{R}^{n_0}$, the one peak random projections and embedding initiation satisfies

$$\mathbb{E}\left[\|a_N W_{N:1}(0)x\|_2^2\right] = \|x\|_2^2.$$

**Proof of Lemma 14** Let $D = W_{p:1}(0)/\sqrt{n_1 n_2 \cdots n_p}$. Then $D$ is an embedding matrix. Thus, $\|Dx\|_2^2 = \|x\|_2^2$. Let $A_i = W_{i:p+1}(0)/\sqrt{n_p n_{p+1} \cdots n_{i-1}}$, where $i \geq p+1$, and $A_p = I$.

Set $B_i = \|A_i Dx\|_2^2 / \|A_{i-1} Dx\|_2^2$, $i \geq p+1$. Then, $B_i$ follows beta distribution $B(n_i/2, (n_{i-1} - n_i)/2)$ given $B_{i-1}, B_{i-2}, \cdots, B_{p+1}$, $i \geq p+1$. If $n_i = n_{i-1}$, then $B_i|(B_{i-1}, B_{i-2}, \cdots, B_{p+1}) = 1$, a.s.

If $B \sim B(a, b)$, then the expectation is given by the following equation,

$$\mathbb{E}B = \frac{a}{a+b}.$$

Thus, by law of total expectation, we have

$$\frac{n_N}{n_p} \mathbb{E} \|a_N W_{N:1}(0)x\|_2^2 = \mathbb{E} \|A_N Dx\|_2^2 = \mathbb{E} B_N B_{N-1} \cdots B_{p+1} \|Dx\|_2^2 = \frac{n_N}{n_p} \|x\|_2^2.$$

This completes the proof. $\qquad\qquad\qquad\qquad\qquad\qquad\qquad\qquad\qquad\qquad\qquad\qquad\qquad\square$

Next, we introduce sub-Gaussian random variables, associated with bounds on how a random variables deviate their expected value.

**Definition 4** *A random variable $X$ with finite mean $\mu = \mathbb{E}X$ is sub-Gaussian if there is a positive number $\sigma$ such that:*

$$\mathbb{E}[\exp(\lambda(X - \mu))] \leq \exp\left(\frac{\lambda^2 \sigma^2}{2}\right) \text{ for all } \lambda \in \mathbb{R}$$

Such a constant $\sigma^2$ is called a proxy variance, and we say that $X$ is $\sigma^2$-sub-Gaussian, and we write $X \sim SG(\sigma^2)$.

**Example 3** *Normal distribution $N(\mu, \sigma^2)$ of course is $\sigma^2$ sub-Gaussian.*
*For beta distribution, Elder (2016) showed that $B(a, b)$ is $\frac{1}{4(a+b)+2}$-sub-Gaussian and later, Marchal & Arbel (2017) concluded $\frac{1}{4(a+b+1)}$-sub-Gaussian.*

The Hoeffding bound for random variable $X$ with mean $\mu$ and sub-Gaussian parameter $\sigma$ is given by,

$$\mathbb{P}[|X - \mu| \geq t] \leq 2 \exp\left\{-\frac{t^2}{2\sigma^2}\right\}, \forall t \geq 0. \tag{51}$$

Simply applying the Chernoff bound for $B(a, b)$, we obtain the following lemma.

**Lemma 15** *Assume random variable $B$ distributed as beta distribution $B(a, b)$ with two positive shape parameters $a$ and $b$. Then*

$$\mathbb{P}(\left|B - \frac{a}{a+b}\right| \geq y) \leq 2 \exp\left\{-2(a+b)y^2\right\}, y \geq 0.$$

*Hence,*

$$\mathbb{P}\left(\left|B - \frac{a}{a+b}\right| \leq \varepsilon \frac{a}{a+b}\right) \geq 1 - \exp\{-\Omega(a^2/(a+b))\},$$

*where $\Omega(\cdot)$ only depend on $\varepsilon$.*
*For the upper tail, we can obtain a better bound,*

$$\mathbb{P}\left(B \geq (1+\varepsilon)\frac{a}{a+b}\right) \leq \exp\left\{-(\varepsilon - \ln(\varepsilon + 1))a\right\}. \tag{52}$$

**Proof of Lemma 15** We only need to prove the third inequality. Assume random variable $B \sim B(a, b)$. Set $v = a+b$, $(1+t)\frac{a}{v} \leq y < 1, t > 0$, and $r > 0$.
We are going to estimate the Chernoff bound for $B$, which is

$$\mathbb{P}(B \geq y) \leq e^{-(ry - \ln \mathbb{E} e^{rB})} =: e^{-I_r(y)}.$$

The moment generating function of $B$ is given by

$$\mathbb{E}e^{rB} = 1 + \sum_{k=1}^{\infty} \frac{a(a+1)\cdots(a+k-1)}{v(v+1)\cdots(v+k-1)} \frac{r^k}{k!} \leq 1 + \sum_{k=1}^{\infty} \frac{a(a+1)\cdots(a+k-1)}{v^k} \frac{r^k}{k!}, r > 0.$$

Recall that the Maclaurin series of $(1 - r/v)^{-a}$ over $(-v, v)$, is given by equation

$$(1 - r/v)^{-a} = 1 + \sum_{k=1}^{\infty} \frac{a(a+1)\cdots(a+k-1)}{v^k} \frac{r^k}{k!}.$$

Thus,

$$I_r(y) = ry - \ln \mathbb{E}e^{rB} \geq ry + a \ln(1 - r/v).$$

Set $r = v - a/y \in (0, v)$. We obtain

$$\mathbb{P}(B \geq y) \leq \exp\{-(vy - a + a\ln(a/(vy)))\} =: \exp\{-vy \cdot g(a/(vy))\}, (1+t)\frac{a}{v} \leq y < 1$$

where $g(x) = 1 - x + x\ln(x)$, $x = a/(vy) \in (0, 1/(1+t)]$. Notice that $g(1) = 0$ and $g'(x) = \ln(x) < 0$ over $x \in (0, 1)$.
We know that

$$g(x) \geq g(1/(1+t)) = \frac{t - \ln(1+t)}{t+1}, t > 0.$$

Thus,

$$\mathbb{P}(B \geq y) \leq \exp\left\{-vy \cdot \frac{t - \ln(1+t)}{t+1}\right\} = \exp\left\{-(t - \ln(1+t))a\right\}, y = (1+t)\frac{a}{v} < 1.$$

Set $y = (1+\varepsilon)\frac{a}{a+b}$. We obtain the inequality (52). $\qquad\square$

**Remark 10** *It is trivial to check*

$$\|W_{j:i}(0)\| = (n_i n_{i+1} \cdots n_j)^{1/2}, 1 \leq i \leq j \leq p,$$
$$\|W_{j:i}(0)\| = (n_{i-1} n_i \cdots n_{j-1})^{1/2}, p+1 \leq i \leq j \leq N,$$
$$\|W_{j:i}(0)\| \leq (n_i n_{i+1} \cdots n_{j-1})^{1/2}(n_p)^{1/2}$$
$$\leq \left(\frac{n_{max}}{n_{min}}\right)^{1/2} (n_i n_{i+1} \cdots n_{j-1} \cdot \max\{n_{i-1}, n_j\})^{1/2}, 1 \leq i < p < j \leq N, (i,j) \neq (1, N).$$

**Remark 11** *As a special case, if $n_1 = n_2 = \cdots = n_{N-1} = n$, we know that $\|W_{j:i}(0)\| = (n_{i-1} n_i \cdots n_{N-1})^{1/2} = n^{(N-i+1)/2}$.*

**Lemma 16** *Assume $n_p/\min\{n_1, n_{N-1}\} \leq C_0 < \infty$. Set $\varepsilon > 0$. Let $C(\varepsilon)$ represent the constant depend only on $\varepsilon$. If $n_1/C_0 \geq C(\varepsilon)n_N$, then with probability at least $1 - e^{-\Omega(n_1/C_0)}$*

$$\sigma_{max}(W_{N:i}(0)) \leq (1+\varepsilon)(n_{i-1} n_i \cdots n_{N-1})^{1/2}, 2 \leq i \leq p$$
$$\sigma_{min}(W_{N:i}(0)) \geq (1-\varepsilon)(n_{i-1} n_i \cdots n_{N-1})^{1/2}, 2 \leq i \leq p.$$

*Similarly, if $n_{N-1}/C_0 \geq C(\varepsilon)rank(X)$, then with probability at least $1 - e^{-\Omega(n_{N-1}/C_0)}$*

$$\sigma_{max}(W_{j:1}(0)|_{\mathcal{R}(X)}) \leq (1+\varepsilon)(n_1 n_2 \cdots n_j)^{1/2}, p+1 \leq j \leq N$$
$$\sigma_{min}(W_{j:1}(0)|_{\mathcal{R}(X)}) \geq (1-\varepsilon)(n_1 n_2 \cdots n_j)^{1/2}, p+1 \leq j \leq N.$$

**Proof of Lemma 16**  Let $D = (n_{N-1}n_{N-2}\cdots n_p)^{-1/2}W_{N:p+1}^T(0)$ and
$A_i = (n_p n_{p-1}\cdots n_i)^{-1/2}W_{p:i}^T(0)$. Assume $v \in S^{n_N-1}$. Easy to see that $A_i$ is a product of random orthogonal projections and $D$ is a random embedding.

Let $e_1 = (1,0,0,\cdots,0)^T \in \mathbb{R}^{n_p}$. There exists orthogonal matrix $T$ such that $TDv = e_1$, $\|e_1\|_2 = \|TDv\|_2 = \|v\|_2 = 1$.

Since random orthogonal projections are right invariant, we have

$$\mathbb{P}(\|A_i Dv\|_2 \geq y) = \mathbb{E}\left[\mathbb{E}\left(I_{\{\|A_i T^T e_1\|_2 \geq y\}}\,\middle|\, D\right)\right] = \mathbb{E}\left[\mathbb{E}\left(I_{\{\|A_i e_1\|_2 \geq y\}}\,\middle|\, D\right)\right] = \mathbb{P}(\|A_i e_1\|_2 \geq y).$$

This proves that $\|A_i Dv\|_2^2$ and $\|A_i e_1\|_2^2$ have the same distribution.

**Claim**: If $v \neq 0$, then $\|A_i Dv\|_2^2 / \|v\|_2^2 = \left\|(n_i n_{i+1}\cdots n_p^2 \cdots n_{N-1})^{-1/2}W_{N:i}^T v\right\|_2^2 / \|v\|_2^2$ follows beta distribution $B(n_{i-1}/2, (n_p - n_{i-1})/2)$.

Define $B_p = \|A_p e_1\|_2^2$, $B_i = \|A_i e_1\|_2^2 / \|A_{i+1} e_1\|_2^2$, $i = p-1, p-2, \cdots, 1$.
Then $B_p \sim B(n_{p-1}/2, (n_p - n_{p-1})/2)$, $B_{p-1}|B_p \sim B(n_{p-2}/2, (n_{p-1} - n_{p-2})/2)$, $\cdots$, $B_i|(B_p, \cdots, B_{i+1}) \sim B(n_{i-1}/2, (n_i - n_{i-1})/2)$.
If $n_{i+1} = n_i$, we know that $B_i|(B_p, \cdots, B_{i+1}) = 1, a.s.$
If $B \sim B(a,b)$, then the moments are given by the following equations,

$$\mathbb{E}B = \frac{a}{a+b}, \text{ and } \mathbb{E}B^k = \frac{a}{a+b}\frac{a+1}{a+b+1}\cdots\frac{a+k-1}{a+b+k-1}.$$

By law of total expectation, we have

$$\mathbb{E}B_i B_{i+1}\cdots B_p = \frac{n_{i-1}}{n_i}\frac{n_i}{n_{i+1}}\cdots\frac{n_{p-1}}{n_p} = \frac{n_{i-1}}{n_p},$$

as well as

$$\mathbb{E}(B_i B_{i+1}\cdots B_p)^k = \frac{n_{i-1}/2}{n_p/2}\frac{n_{i-1}/2+1}{n_p/2+1}\cdots\frac{n_{i-1}/2+k-1}{n_p/2+k-1}.$$

Notice that all integer moments of $B_i B_{i+1}\cdots B_p$ match those of $B(n_{i-1}/2, (n_p - n_{i-1})/2)$. We can verify that beta distribution satisfies Carleman's condition, which implies that $B_i B_{i+1}\cdots B_p \sim B(n_{i-1}/2, (n_p-n_{i-1})/2)$.

Thus, $\|A_i Dv\|_2^2 / \|v\|_2^2 \sim B(n_{i-1}/2, (n_p - n_{i-1})/2)$, which proves the claim.

With probability at least $1 - \exp\{-\Omega(n_1/C_0)\}$, we have

$$(1-\varepsilon)^2\frac{n_{i-1}}{n_p} \leq \|ADv\|_2^2 \leq (1+\varepsilon)^2\frac{n_{i-1}}{n_p}, \|v\|_2 = 1.$$

Using the $\phi-$net technique which has already been used to prove lemma 11, we know that

$$\sigma_{min}(AD) \geq (1-\varepsilon)\left(\frac{n_{i-1}}{n_p}\right)^{1/2},$$

and

$$\sigma_{max}(AD) \leq (1+\varepsilon)\left(\frac{n_{i-1}}{n_p}\right)^{1/2},$$

with probability at least $1 - \exp\{n_N \ln(3/\phi(\varepsilon))\}\exp\{-\Omega(n_1/C_0)\} \geq 1 - \exp\{-\Omega(n_1/C_0)$, since $n_1/C_0 \geq C(\varepsilon)n_N$, for $2 \leq i \leq p$.
Hence, with probability at least $1 - e^{-\Omega(n_1/C_0)}$, we have

$$\sigma_{min}(W_{N:i}(0)) \geq (1-\varepsilon)(n_{i-1}\cdots n_{N-1})^{1/2},$$

and

$$\sigma_{max}(W_{N:i}(0)) \leq (1+\varepsilon)(n_{i-1}\cdots n_{N-1})^{1/2}.$$

The other part of the proof is similar to that of lemma 12, so we omit it.

$\square$

**Proof of Theorem 5** Set $c > 0$, $c_1 = c/6, c_2 = c/3$. In lemma 16, we can pick a $\varepsilon > 0$, such that $1 + \varepsilon \leq e^{c_1/2}$ and $1 - \varepsilon \geq e^{-c_2/2}$. Set $M = 2\sqrt{C_0}, \theta = 0$, $B_0 = B_\delta$, and $\eta = \frac{(1-\varepsilon)2n_N}{e^{2c}\beta N}$.

The requirement on size $\{n_1, n_2, \cdots, n_{N-1}, N\}$ in (24) make sure that the remark 10, lemma 16, lemma 2, and lemma 3 all hold.

Notice that even though we need the conclusions in lemma 16 simultaneously hold for $2 \leq i \leq p$, $p + 1 \leq j \leq N$, it suffices to apply lemma 16 over $i \in I$ and $j \in J$, such that $\{n_i; i \in I\}$ and $\{n_j; j \in I\}$ both have distinct values. Since $|I| \leq \underline{N}$ and $|J| \leq \underline{N}$, with probability at least

$$1 - 2\underline{N}e^{-\Omega(n_{min}/C_0)} - \delta/2 \geq 1 - \delta,$$

the one peak random orthogonal projections and embeddings initialization satisfies the initialization assumption (14) and the overparameterization assumption (15).

Under assumption $n_1 = n_2 = \cdots = n_{N-1}$, we can use remark 11 to replace lemma 16. Thus, with probability at least $1 - \delta/2 \geq 1 - \delta$, (14) holds. Applying lemma 2 and 3, we complete the proof. □

**Proof of Theorem 6** Let $W_N(0) = \sqrt{n}U_N[I_{n_y}, 0]V_N^T, \cdots, W_i(0) = \sqrt{n}U_i I_n V_i^T, 2 \leq i \leq N - 1$, and $W_1(0) = \sqrt{n}U_1[I_{n_x}, 0]^T V_1^T$. Now, we want to verify (14). By simply calculation, we have

$$\begin{cases} \sigma_{max}(W_{N:i+1}(0)) = \sigma_{min}(W_{N:i+1}(0)) = n^{(N-i)/2}, 1 \leq i \leq N - 1, \\ \sigma_{max}(W_{i-1:1}(0)|_{\mathcal{R}(X)}) = \sigma_{max}(W_{i-1:1}(0)|_{\mathcal{R}(X)}) = n^{(i-1)/2}, 2 \leq i \leq N, \\ \|W_{j:i}(0)\| = n^{(j-i+1)/2}, 1 < i \leq j < N. \end{cases}$$

Notice that for any $1 \leq p \leq m$

$$\|a_N W_{N:1}(0)x\|_2^2 = \frac{n}{n_N}\left\|U_N[I_{n_y}, 0]V_N^T U_N[I_{n_x}, 0]^T V_1^T x\right\|_2^2 = \frac{n}{n_N}\left\|U_N[I_{n_y}, 0]V_N^T x'\right\|_2^2,$$

where $x' = U_N[I_{n_x}, 0]^T V_1^T x$, $\|x\|_2 = \|x'\|_2$.

Since the distribution of $U_N[I_{n_y}, 0]V_N^T$ is right invariant under multiplying orthogonal matrices, we have

$$\left\|U_N[I_{n_y}, 0]V_N^T x'\right\|_2^2 / \|x\|_2^2 \sim B(\frac{n_y}{2}, \frac{n - n_y}{2}).$$

Thus,

$$\mathbb{E}\left[\|a_N W_{N:1}(0)x\|_2^2\right] = \|x\|_2^2.$$

Applying lemma 2, we have

$$L_0 - L(W_*) \leq \beta\left(\frac{2 \cdot rank(X)}{\delta} + \|W_*\|_X^2\right),$$

with probability at least $1 - \delta/2$.

Applying Lemma 3 with $c > 0$, $c_1 = c/6, c_2 = c/3$, $\theta = 0$, we complete the proof. □

**Proof of Theorem 1** Theorem 1 is a special case of Theorem 4 and Theorem 5. Hence, we omit the proof. □

**Proof of Theorem 2** In Theorem 4, 5, and 6, we proved that for given constant $c_1, c_2 > 0$ and $0 < \varepsilon, \delta/2 < 1/2$ as well as learning rate $\eta$, there exists constant $C = C(c_1, c_2)$ such that all three kinds of random initializations will fall into the convergence region defined in Section 4.1 with probability at least $1 - \delta$.

This implies that with probability at least $1 - \delta$, we obtain (16) by Lemma 4. Applying Lemma 6, we complete the proof. □

# G    Numerical Experiments

We will discuss some empirical evidence to support the main results in Section 3. We aim to show how the trajectories of the non-convex deep linear neural networks are related to a convex optimization problem for GD under different initialization schemes. Consider the following procedures for plots of the logarithm of loss as a function of number of iterations:

a) We choose $X \in \mathbb{R}^{128 \times 1000}$ and $W_* \in \mathbb{R}^{10 \times 128}$ and set $Y = W_* X + \varepsilon$, where the entries in $X$, $W_*$ and $\varepsilon$ are drawn i.i.d. from $N(0, 1)$.

b) We consider the loss function $\frac{1}{2} \left\| a_N W_{N:1} X - Y \right\|_F^2$.

c) For the given linear networks, we apply the Gaussian initialization and the one peak random orthogonal projections and embeddings initialization, which are denoted as $W_j(0), 1 \le j \le N = 3$.

d) For the convex optimization problem (1), we set the initialization to be $W(0) = a_N W_N(0) \cdots W_1(0)$.

e) We set the learning rate $\eta = \frac{n_N}{N \cdot \|X\|^2}$ and $\eta_* = \frac{N}{n_N} \eta$ for the deep linear neural networks and the convex optimization problem, respectively.

f) We draw the loss function through 25 iterations.

Figure 1 compare the trajectories of the logarithm of loss for gradient descent in deep linear networks and the corresponding convex optimization problem. The left panel shows the comparison with Gaussian initialization, while the right panel shows the comparison with orthogonal initialization, both without averaging.

Recall that the main theorems in Section 3 require $n_{min} \ge \max\{n_0, n_N\}$, and thus, in the simulation, we require $n_{min} \ge 128$. Thus, for the top panels, the minimal width of the hidden layers $n_{min} = 128$ is small, and the trajectories of loss for deep linear networks exhibit non-monotonic behavior, with the loss increasing in some iterations. However, this does not contradict the main theorems in Section 3, because the minimal width of the hidden layers is insufficient.

As the minimal width of the hidden layers increases, the trajectories of loss for deep linear networks become increasingly similar to those of the corresponding convex optimization problem. This suggests that increasing the minimal width of the hidden layers helps to stabilize the optimization process and makes it more closely resemble the convex counterpart, which is consistent with Theorem 2.

The choice of initialization scheme, whether Gaussian or orthogonal, does not significantly impact the overall convergence behavior and the relationship between the trajectories of deep linear networks and the convex optimization problem, when the minimal width of the hidden layers is large enough.

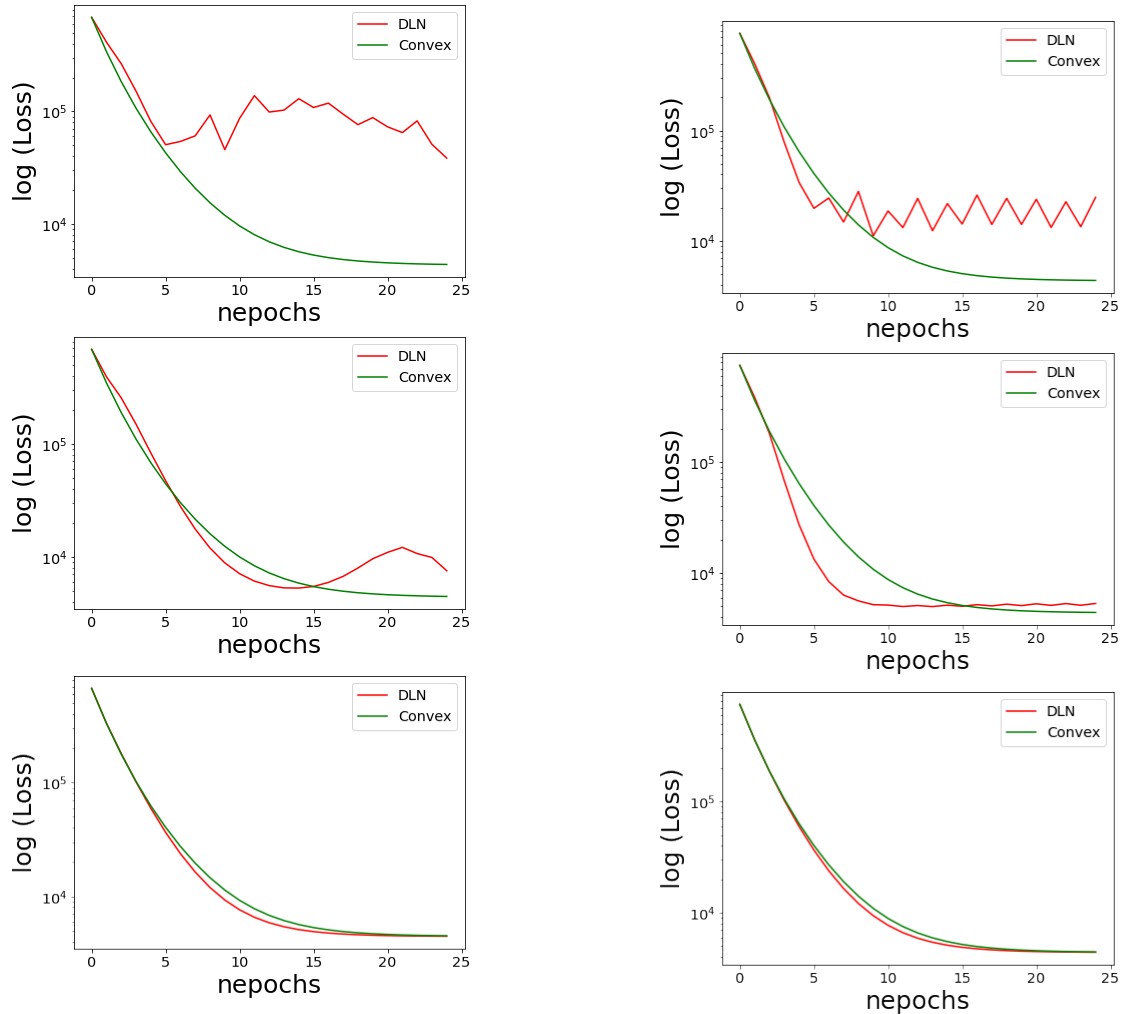

Figure 1: Plot of logarithm of loss as a function of number of iterations with $n_1 = n_2 = n_3 = 128$ (Top), 200 (Middle), 2000 (Bottom) for Gaussian initialization (left panel) and $n_1 = n_2 = n_3 = 128$ (Top), 200 (Middle), 5000 (Bottom) for Orthogonal initialization (right panel), respectively.

