# OpenReview forum: "Convergence Analysis and Trajectory Comparison of Gradient Descent for Overparameterized Deep Linear Networks"
_TMLR — Accepted by TMLR_

### Review · Reviewer_H61G · 2024-04-23

**Summary Of Contributions:**

The authors have analyzed the set of initializations which give rise to linear convergence of gradient descent (GD). Then they obtain the minimum width that is sufficient for GD with random initialization to converge with high probability.

The authors claim that the most significant contribution and novelty of this work is to obtain trajectory comparison results and show that the trajectories of gradient descent for such networks can be made arbitrarily close to those of the original convex optimization problem.

**Audience:**

Yes

**Broader Impact Concerns:**

This is a theoretical work. There is no such concern.

**Claims And Evidence:**

No

**Requested Changes:**

Please see weaknesses and address the raised concerns.

**Strengths And Weaknesses:**

Providing a detailed proof sketch and clearly citing the relevant results when establishing theoretical results are among the strengths of the paper. This paper is overall well-written.

-----------------------

Weaknesses:

$l$ is not defined in (1). Do you consider all convex losses or specific loss functions?

The strong convexity and smoothness assumptions are stated later on page 3. This creates confusion as the reader does not know the problem setup clearly when reading (1).

-----------------------

One page 3, the authors claimed "Notice that $W_*$ might not be unique, but $W_* P_X$ is unique." This is not really clear. Please provide the proof.

-----------------------

The claim that GD in (7) is equivalent to that of (9) in Remark 1 is unclear specially since $\nabla L$ is not necessarily a homogeneous function.

-----------------------

To help the readers and improve the presentation of the paper, the authors are encouraged to elaborate on the definitions of quantities in Eqs (10) and (11). In particular, what each expression in Eq. (11) means?

-----------------------

The sufficient overparameterizations in Theorem 1 (both parts 1 and 2) are not clear. The LHS and RHS of the inequalities depend on   $n_{\min}$. In particular, the RHS depends on $n_{\min}$ through $C_0$ which is an upper bound on the ratio $\frac{n_{\max}}{n_{\min}}$.

The same issue holds for Theorem 2.

-----------------------


The term $q+\tau$  in (12c) can be larger than 1. Then, the exponential growth with $t$ will not support the claim that is stated just above Theorem 2 and introduction regarding the closeness of trajectories.

Based on the arguments in Corollary 3, the authors adjust $\tau$ as they wish. But then what is the cost of scaling down $\tau$ in terms of convergence rate and/or the required assumptions?

-----------------------

The numerical results in Appendix G hold under different learning rates for the deep linear neural networks and convex optimization problem, which are not consistent with theoretical results. It is unclear the role of learning rates on the claims regarding closeness of trajectories.

-----------------------



The constant learning rate significantly limits the trajectory of optimization. In practice, time-varying and adaptive  learning rates are quite popular. The reviewer is wondering  why the authors focused only on fixed learning rates for example in all GD updates?

-----------------------

On page 2: "unless the width is almost linear.":  please clarify linear in what in "unless the width is almost linear"

---

> ### Author Response · Authors · 2024-04-30
> **Response to Reviewer: Addressing Requested Changes and Providing Clarifications**
>
> We sincerely thank the Reviewer for their thorough and insightful review. Their constructive feedback and valuable suggestions have helped us identify areas for clarification and improvement, and prompted us to consider alternative perspectives. Addressing these points will significantly enhance the quality and impact of our work. We appreciate the Reviewer's expertise and dedication in evaluating our research.  In response to the Reviewer's comments, we have made the following changes and clarifications:
> 1. Description for $l$ in (1) has been added: $l(\cdot,y)$ is strongly convex and has a smooth gradient with respect to its first argument uniformly in $y$.
> 2. The explanation of the statement "$W_*P_X$ is unique" has been added. The proof of this statement can be found in Lemma 7.
> 3. Remark 1 essentially addresses the "Scaling of difference equations." Therefore, a homogeneous assumption of $\nabla L$ is not required.
> 4. Here, the quantity $q$ is related to the rate of convergence of gradient descent for the strongly convex function $L(W)$. Setting the learning rate $\eta_*=\frac{2}{\alpha+\beta}$, we can rewrite the well-known rate in equation (4) in terms of $q$, due to
> \begin{equation*}
>     q=1-\alpha \eta_* (2-\eta_* \alpha)=1-\frac{ 4\kappa }{ (1+\kappa)^2 }.
> \end{equation*}
>
> The parameter $\delta\in (0,\frac{1}{2})$ is a confidence threshold, such that the conclusions in Theorem 1 (or Theorem 2) hold with a probability of at least $1-\delta$, given an extra inequality constraint between the minimum width of hidden layers, $n_{min}$, and the depth, $N$, as quantified by $\mathbb{C}_1,\mathbb{C}_2,\mathbb{C}_3.$
>
> 5. In our analysis, we require that the ratio $ n_{\text{max}}/n_{\text{min}} $ be upper bounded by a pre-specified constant $ C_0 $ for networks with varying width. In Du \& Hu (2019) and Hu et al. (2020), $ C_0 = 1 $. Throughout the paper, we stipulate that for the deep linear network, $ n_{\text{max}}/n_{\text{min}} \leq C_0 $. The effect of $ C_0 $, $\delta$ and information for networks are quantified by the quantities $ \mathbb{C}_1, \mathbb{C}_2, \mathbb{C}_3 $.
> 6. Due to $q$ being the quantity that only depends on the convex loss function $L(W)$, and $\tau$ can be chosen arbitrarily, we can always require that $q+\tau < 1$. As $\tau$ goes to $0$, only the constant $C( \tau ,\eta/\eta_1)$ will increase, which is the only cost of scaling down $\tau$. Alternatively, one can consider that for a sufficiently large $n_{min}$, $\tau$ can be selected to be sufficiently small.
> 7. Recall that the main theorems in Section 3 require $n_{min}\geq \max {n_0, n_N}$, and thus, in the simulation, we have $n_{min}\geq 128$. Thus, for the top panels, the minimal width of the hidden layers $n_{min}=128$ is small, and the trajectories of loss for deep linear networks exhibit non-monotonic behavior, with the loss increasing in some iterations. However, this does not contradict the main theorems in Section 3, as the minimal width of the hidden layers is insufficient.   As the minimal width of the hidden layers increases, the trajectories of loss for deep linear networks become increasingly similar to those of the corresponding convex optimization problem. This suggests that increasing the minimal width of the hidden layers helps to stabilize the optimization process and makes it more closely resemble the convex counterpart, which is consistent with Theorem 2. In principle, the learning rates, which are smaller that $\eta_1$, do not affect the closeness of trajectories result stated in Theorem 2. Only the overparameterization condition will affect the closeness of trajectories result in Theorem 2.
> 8. With some effort, our proof can potentially be adapted to accommodate time-varying and adaptive learning rates, as long as $\sum_{i=1}^{\infty} \eta_i = \infty$, and all $\eta_i$ are less than $(1-e')\eta_{max}$ for some $e' \in (0,1)$. Additionally, the condition $\sum_{s=1}^\infty \eta_s \prod_{i=1}^s (1-\eta_i \gamma_i )^{1/2}<\infty$ is required to make the proof of Claim 1 valid. For example, learning rates $\eta_s = O\left(\frac{1}{s}\right)$ do satisfy this condition. Here, the term $\prod_{i=1}^s (1-\eta_i \gamma_i )$ is associated with the decay rate of the loss, where $L_s - L_0 \leq \prod_{i=1}^s (1-\eta_i \gamma_i) B$.
> 9. Hu et al. (2020) established that the convergence for Gaussian initialization can be very slow for deep linear neural networks with large depths. For efficient convergence of Gaussian networks, the width needs to scale linearly with the depth.

---

> > ### Comment · Reviewer_H61G · 2024-06-11
> > **Response to Authors**
> >
> > I would like to thank the authors for their response. While some issues are addressed in the response and revision, there are still some inaccurate claims that are not addressed after reading the authors' response:
> >
> > The claim that GD in (7) is equivalent to that of (9) in Remark 1 is unclear specially since $\nabla L$ is not necessarily a homogeneous function.
> >
> > ------------------
> >
> > The sufficient overparameterizations in Theorem 1 (both parts 1 and 2) are not clear. The LHS and RHS of the inequalities depend on $n_{\min}$. In particular, the RHS depends on $n_{\min}$ through $C_0$ which is an upper bound on the ratio $\frac{n_{\max}}{n_{\min}}$.
> >
> > The same issue holds for Theorem 2.

---

> > > ### Author Response · Authors · 2024-06-12
> > > **Brief proof of Remark 1 and Explanation of overparameterization condition**
> > >
> > > Brief proof of Remark 1:
> > >
> > > First of all, equation (7) is the GD update for (6), i.e.,
> > > \begin{equation}
> > > W_j(t+1)=W_j(t)-\eta \frac{ \partial L(a_NW_{N:1}(t)) }{\partial W_j(t)  }.
> > > \end{equation}
> > > Consider the scaling $V_i(t)=\frac{1}{\sqrt{n_i}}W_i(t)$. Thus $d V_i(t) =\frac{1}{\sqrt{n_i}}dW_i(t)$ and $ L(a_NW_{N:1}(t))=L(V_{N:1}(t))$. Then
> > > \begin{equation}
> > > V_j(t+1)=\frac{1}{\sqrt{n_j}}W_j(t+1)=\frac{1}{\sqrt{n_j}}W_j(t) -\eta \frac{1}{\sqrt{n_j}}\frac{ \partial L(a_NW_{N:1}(t)) }{\partial W_j(t)  }=V_j(t) -\frac{\eta }{ {n_j}}\frac{ \partial L(V_{N:1}(t)) }{\partial V_j(t)  }=V_j(t) -\frac{\eta }{ {n_j }} (V_{N:j+1}(t))^T { \nabla L(V_{N:1}(t)) }  (V_{j-1:1}(t))^T.
> > > \end{equation}
> > >
> > > Explanation of overparameterization condition:
> > >
> > > Here, I encourage two different ways to understand the overparameterization condition in Theorem 1 and 2.
> > >
> > > a) We treat $C_0$ as a constant specified by the user, which is independent of the network structure. All networks considered in Theorem 1 and Theorem 2 must satisfy that $n_{min}\geq max (n_0,n_N) $ and $n_{max}/n_{min}\leq C_0$ as the default condition, which is mentioned at the beginning to section 3.
> > >
> > > Now, if we consider $C_0=1$, i.e., $n_{min}=n_{max}$, then our theorem 1 essentially recovers the main theorem in Du & Hu (2019), and Hu et al. (2020).
> > >
> > > b) For a given network structure, we can set $C_0=n_{max}/n_{min}$. Now, $C_0$ depends on $n_{min}$ and $n_{max}$. Due to only quantities in (11) depending on $C_0=n_{max}/n_{min}$, we can rewrite the first overparameterization condition in Theorem 2 as
> > > \begin{equation}
> > > n_{min}\geq C (n_N \kappa^2 B_\delta\frac{n_{max}}{n_{min}} +\ln N )  N \iff n^2_{min}\geq C (n_N \kappa^2 B_\delta {n_{max}}   + \ln N \cdot n_{min} )  N.
> > > \end{equation}
> > > However, the above new overparameterization condition, which eliminates $C_0$, looks more complicated.

---

### Review · Reviewer_dezf · 2024-05-02

**Summary Of Contributions:**

The paper contributes to the understanding of the efficiency of gradient descent (GD) in training deep neural networks. It focuses on overparameterized deep linear neural networks and compares the GD trajectory with that of the corresponding convex optimization problem. The results demonstrate a close match between the two trajectories, highlighting the effectiveness of GD in practice. The study addresses a significant theoretical problem in machine learning and sheds light on why GD methods are efficient for practical applications. Additionally, the paper reveals that the GD trajectory for overparameterized deep linear networks avoids bad saddle points, further enhancing its effectiveness.

**Audience:**

Yes

**Claims And Evidence:**

Yes

**Requested Changes:**

1. Is it possible to achieve similar results using stochastic gradient descent (SGD)? What are the main difficulties?

2. What will happen when the width is fixed but the depth of the linear neural network is sufficiently large? Additionally, what happens when both the width and depth of the network vary?

3. The assumption $n_{min} \ge \max(n_0, n_N)$ is kind of stringent, leading to results that may not be applicable to high-dimensional data. Any remarks on this limitation?

**Strengths And Weaknesses:**

Strengths:
1. The paper presents a thorough convergence analysis and trajectory comparison of the gradient descent method for overparameterized deep linear neural networks.
2. The paper highlights the importance of appropriate initialization and sufficient width of hidden layers for achieving close alignment between the trajectories of gradient descent and the convex optimization problem.

Weaknesses:
1.  The paper primarily focuses on overparameterized deep linear neural networks. While this provides valuable insights into this specific domain, the findings may not directly extend to more general deep neural network architectures.

---

> ### Author Response · Authors · 2024-05-05
> **Response to Reviewer Comments**
>
> We greatly appreciate the reviewer's recognition of the paper's strengths and the valuable points you have raised. Below, we address your requested changes and provide additional clarifications.
>
> 1. Stochastic Gradient Descent (SGD): While our paper primarily focuses on the analysis of gradient descent (GD), extending these results to include stochastic gradient descent (SGD) represents an important direction for future research. The main challenge in analyzing SGD is the stochastic nature of the gradients, which introduces additional noise and variability that can hinder the rate of convergence. To achieve results comparable to those for GD, a careful analysis of SGD's convergence behavior is required, taking into account factors such as batch size and learning rate schedule. This will likely necessitate appropriate modifications to Lemma 3.
>
> 2. Fixed Width and Varying Depth: Thank you for your question regarding the impact of fixed width and varying depth on the convergence behavior of deep linear neural networks. In our paper, we have addressed this aspect to some extent. Our overparameterization conditions in Theorem 1 and Theorem 2 align with the findings of Hu et al. (2020), showing that for orthogonal initializations, the width needed for efficient convergence to a global minimum is independent of the depth. In contrast, for Gaussian initializations, the required width scales linearly with the depth. Regarding the scenario where both the width and depth of the network vary, our Theorem 1 and Theorem 2 provide guarantees for the convergence behavior and trajectory comparison within the regime of overparameterization. However, when the width and depth do not fall into the regime of overparameterization, the optimization dynamics become more complex.
>
> 3. In the literature on overparameterized linear networks, both Du & Hu (2019) and Hu et al. (2020) required $n_{min} \geq \max(n_0, n_N),$ and they showed linear convergence of gradient descent (GD). For convergence analysis of non-overparameterized linear networks, Arora et al. (2019) required a more natural condition, $n_{min} \geq \min(n_0, n_N)$. Note that $n_{min} \geq \min(n_0, n_N)$ holds if and only if, for any $W_*$, there exist $W_1, \cdots, W_N$ such that $W_* = W_N \cdots W_1$. However, to show linear convergence, it requires an extra deficiency margin assumption, which generally leads to a very small convergence region for initialization.
>
> 4. We might be able to based on neural tangent kernel, to extend our work to compared the GD trajectory of general deep neural network architectures for large width, with to its linearization.
>
> Du, S., & Hu, W. (2019). Width provably matters in optimization for deep linear neural networks. In International Conference on Machine Learning  (ICML).
>
> Hu, W., Xiao, L., & Pennington, J. (2020). Provable Benefit of Orthogonal Initialization in Optimizing Deep Linear Networks. In International Conference on Learning Representations (ICLR).
>
> Arora, S., Cohen, N., Golowich, N., & Hu, W. (2019). A convergence analysis of gradient descent for deep linear neural networks. In International Conference on Learning Representations (ICLR).

---

### Review · Reviewer_BJ3T · 2024-05-07

**Summary Of Contributions:**

This paper presents an analysis of gradient descent (GD) in deep linear networks and provides two new theoretical results:
1. **Convergence of Gradient Descent**: The authors demonstrate that, for deep linear networks with a strongly convex loss function and sufficient width, GD with appropriate initialization converges linearly with high probability. This result extends previous work that primarily focused on $l_2$ loss.
2. **Trajectory Approximation**: For sufficiently wide deep linear networks, the trajectory of gradient descent closely resembles that of a simplified convex problem.

**Audience:**

No

**Claims And Evidence:**

Yes

**Requested Changes:**

To enhance the practical impact of the results, the authors may want to justify how the techniques and/or insights can be extended to deep nonlinear networks (not just simply claiming ``can potentially extend to more general deep neural network architectures''.

**Strengths And Weaknesses:**

## Strength
1. **Novel Analysis of GD Trajectory**: It appears that this paper offers the first comprehensive comparison of gradient descent (GD) trajectories in deep linear networks with those of simplified convex problems.
2. **Rigorous and Clear Proof Framework**: The proof is robust, and the framework is well-structured. It first establishes sufficient conditions for convergence by characterizing the convergence region, then demonstrates how various initializations fall into this region with high probability as network width increases.

## Weakness
My main concern is whether the insights and techniques derived in this paper can carry over to deep nonlinear networks. In practice, deep linear networks are rarely used. One major goal of studying deep linear networks is to provide insights for understanding non-linear networks.
1. **Difference in Landscape**: Recent research demonstrated that the loss landscapes of non-linear networks differ significantly from those of linear networks. For instance, spurious local minima commonly occur in deep non-linear networks [R1, R2], whereas deep linear networks do not exhibit spurious local minima. This discrepancy suggests that the training dynamics of non-linear networks are unlikely to closely resemble those of linear networks.
2. **Recent advances on NTK**: Recent analyses based on the Neural Tangent Kernel (NTK) have characterized the training dynamics of sufficiently wide non-linear networks. It remains unclear how the analysis of deep linear networks can provide deeper insights into the training dynamics of non-linear networks.

[R1] Tian Ding, Dawei Li, and Ruoyu Sun. "Suboptimal local minima exist for wide neural networks with smooth activations." Mathematics of Operations Research 47.4 (2022): 2784-2814.

[R2] Fengxiang He, Bohan Wang, and Dacheng Tao. "Piecewise linear activations substantially shape the loss surfaces of neural networks." International Conference on Learning Representations. 2019.

---

> ### Author Response · Authors · 2024-05-10
> **Insights from Deep Linear Networks and Their Potential Extension to Non-Linear Networks**
>
> We appreciate your insightful comments and the opportunity to address your concerns regarding the applicability of our findings to deep non-linear networks. We understand the importance of extending the insights gained from studying deep linear networks to the more practical setting of non-linear networks. In this response, we aim to provide a perspective on how our work can contribute to understanding the optimization dynamics of wide neural networks.
>
> First of all, it is worth noting that recent work [R3] has already demonstrated that gradient descent finds a global minimum in training for sufficiently wide deep neural networks, including multilayer fully-connected neural networks, ResNet, and convolutional ResNet. The linear rate of convergence in terms of loss has been established in these cases. However, the trajectory comparison results has not been established.
>
> To address your concerns regarding the existence of bad local minima in the landscape and the linear convergence of gradient descent for wide neural networks, let us first offer some insights from our analysis of wide linear networks:
>
> 1. Landscape: In linear networks, saddle points exist, making it challenging to establish linear convergence for every initial weight.
> 2. Convergence region: We have identified a convergence region such that if the initial weight matrices fall into this region, linear convergence can be guaranteed.
> 3. High probability property for convergence region with classical random initialization schemes: We have shown that if the network is sufficiently wide, Gaussian and orthogonal initializations will fall into the convergence region with high probability.
> 4. Impact of saddle points: The existing saddle points do not hinder the optimization of wide linear networks because there are no saddle points in the high-probability convergence region.
>
> Building upon these insights, we propose the following thoughts for non-linear networks:
> 1. Landscape: As highlighted in [R1, R2], spurious local minima commonly occur in deep non-linear networks. To ensure linear convergence, the initial weights must be away from saddles and the "convergence region of suboptimal minimizers."
> 2. High probability property for convergence region (for global minimizer) with classical random initialization schemes: We believe that a convergence region for wide non-linear networks must exist, given the established linear convergence results in [R3].
> 3. Trajectory comparison candidate: The Neural Tangent Kernel (NTK) could potentially serve as a candidate for trajectory comparison in the non-linear setting.
> 4. Impact of spurious local minima: The existence of spurious local minima may not necessarily hinder the optimization of wide networks if the initialization falls into the "convergence region of suboptimal minimizers" with low probability.
>
> Our understanding and intuition are as follows:
>
> Pure landscape analysis does not incorporate the probability structure introduced by random initialization. If we combine landscape analysis with random initialization, we might be able to claim the following statement: as the width of the network increases, the probability of random initialization falling into the convergence region of nonlinear networks (for a global minimizer) approaches 1.
>
> We believe that our work on deep linear networks provides a foundation for understanding the optimization dynamics of wide networks and can guide future research in the non-linear setting. By characterizing convergence regions, analyzing the properties of initialization schemes, and leveraging insights from NTK-based approaches, we aim to bridge the gap between the theoretical understanding of deep linear networks and the practical challenges of training deep non-linear networks in our future work.
>
> [R1] Tian Ding, Dawei Li, and Ruoyu Sun. "Suboptimal local minima exist for wide neural networks with smooth activations." Mathematics of Operations Research 47.4 (2022): 2784-2814.
>
> [R2] Fengxiang He, Bohan Wang, and Dacheng Tao. "Piecewise linear activations substantially shape the loss surfaces of neural networks." International Conference on Learning Representations. 2019.
>
> [R3] Du, S., Lee, J., Li, H., Wang, L. &amp; Zhai, X.. (2019). Gradient Descent Finds Global Minima of Deep Neural Networks. International Conference on Machine Learning

---

### Comment · Action_Editor_n3et · 2024-06-16
**Requests**

Dear Authors,

The proof of Theorem 2 is quite brief and I find it difficult to follow. Theorem 4-6 might imply the claim in Eq. (12c), but I'm not sure if it is obvious for those in Eqs. (12b) and (12c). Could you add more details in the proof?

I also request the authors to provide a proof or a citation for Eqs. (3-4) and give references to the lemmas for the claim just below Eq. (8) (I'm assuming Lemma 9 and Lemma 14 are relevant here).

Where can I find the definition of $\mathcal{R}(X)$? What condition does "falling in the convergence region" refer to? I do see some informal explanations, but I'm not sure if there is a formal definition in the manuscript.

---

> ### Author Response · Authors · 2024-06-16
> **Response to Action Editor Comments**
>
> We appreciate AE's insightful comments. We have modified the manuscript accordingly.
>
> $\textbf{Concerns related to proof of Theorem 2}$:
>
> First of all, Eqs. (12a), (12b) and (12c) are direct consequences of Lemma 6.
>
> There are 3 steps in proving Theorem 2.
>
> Step 1: Show random initialization falls into the convergence region, which is related to Theorem 4,5 and 6.
>
> Step 2: We only need the result that Eq. (16) holds with probability at least $1-\delta$.
>
> Step 3: Apply Lemma 6.
>
> $\textbf{Citation for Eqs. (3-4)}$:
>
> Eqs. (3) and (4) are essentially the boundary special cases for Theorem 3.9 in [R1] and Theorem 2.1.15 in [R2].
>
> $\textbf{References for Lemmas for the the claim just below Eq. (8)}$:
>
> References for Lemmas 9 and 14 have been added.
>
> $\textbf{Definition of}$ $\mathcal{R}(X)$:
>
> $\mathcal{R}(X)$ is defined in section 4.1.
>
>
> $\textbf{Convergence Region}$:
>
> The definition of the convergence region can be found in Section 4.1, Definition 2.
>
> [R1] Garrigos, Guillaume, and Robert M. Gower. "Handbook of convergence theorems for (stochastic) gradient methods." arXiv preprint arXiv:2301.11235 (2023).
>
> [R2] Nesterov, Yurii. Introductory lectures on convex optimization: A basic course. Vol. 87. Springer Science & Business Media, 2003.

---

### Decision · Action_Editor_n3et · 2024-06-29

**Recommendation:** Accept with minor revision

**Comment:**

This is a borderline paper for which the reviewers had negative opinions about its relevance (see my comments in the Audience section). However, the results provide some insights about over-parametrization and the claims are supported by rigorous proofs. Hence, the quality of the paper does not fall below the acceptance bar of TMLR according to [its acceptance criteria](https://jmlr.org/tmlr/acceptance-criteria.html). I (weakly) recommend acceptance.

On the other hand, the contributions of this work may not be sufficient to receive any special _certifications_.

**Audience:**

As two reviewers mentioned, the relevance of this work is limited because it studies deep _linear_ networks, which in practice are rarely preferred over deep _non-linear_ networks. A reviewer also pointed out that recent studies (Ding et al., 2022; He et al., 2019) show that the loss landscape will be significantly different when there is non-linearity, and thus it is difficult to find a reason to study deep linear networks.

There was a comment on the interpretability of the over-parametrization conditions, which partially remains unclear.

Nevertheless, two reviewers expressed that the results may be interesting or insightful for researchers working on a specific domain (such as ones studying the nature of over-parametrization).

**Claims And Evidence:**

This paper studies the gradient descent method when applied to strongly convex loss functions with over-parametrized deep linear neural networks.
For networks with sufficiently wide hidden layers, initialized with any of the three initialization methods mentioned in the paper, the authors show that the convergence and the trajectory of the gradient descent is close to the corresponding convex optimization problem (i.e., that of the linear network with no hidden layer). The results generalize those of Du & Hu (2019), and Hu et al. (2020) to more general, strongly convex loss functions, with the novel study on the trajectory. The authors provide complete proofs for their claims and a few simple simulation results. There were some clarity issues raised during the discussion period, but most of them have been addressed by the authors' responses.